# MACCA: Offline Multi-agent Reinforcement Learning with Causal Credit Assignment

## Abstract

Offline Multi-agent Reinforcement Learning (MARL) is valuable in scenarios where online interaction is impractical or risky. While independent learning in MARL offers flexibility and scalability, accurately assigning credit to individual agents in offline settings poses challenges due to partial observability and emergent behavior. Directly transferring the online credit assignment method to offline settings results in suboptimal outcomes due to the absence of real-time feedback and intricate agent interactions. Our approach, MACCA, characterizing the generative process as a Dynamic Bayesian Network, captures relationships between environmental variables, states, actions, and rewards. Estimating this model on offline data, MACCA can learn each agent's contribution by analyzing the causal relationship of their individual rewards, ensuring accurate and interpretable credit assignment. Additionally, the modularity of our approach allows it to seamlessly integrate with various offline MARL methods. Theoretically, we proved that under the setting of the offline dataset, the underlying causal structure and the function for generating the individual rewards of agents are identifiable, which laid the foundation for the correctness of our modeling. Experimentally, we tested MACCA in three environments, including discrete and continuous action settings. The results show that MACCA outperforms SOTA methods and improves performance upon their backbones.

## 1 Introduction

Offline Reinforcement learning (RL) has gained significant popularity in recent years. It can be particularly valuable in situations where online interaction is impractical or infeasible, such as the high cost of data collection or the potential danger involved (Jiang & Lu, 2021; Meng et al., 2021; Zhang et al., 2023). Working along this line, Multi-agent Reinforcement Learning (MARL) further extends its applicability to variable domains, such as autonomous driving (Shi et al., 2021; Florbäck et al., 2016; Codevilla et al., 2018), robotics coordination (Zhang & Ouyang, 2012; Chiddarwar & Babu, 2011), and resource allocation (Tesauro et al., 2007; 2006). The independent learning paradigm in MARL is appealing due to its flexibility and scalability, making it a promising approach to solving complex problems in dynamic environments (de Witt et al., 2020; Lyu et al., 2021).

While independent learning in MARL has its merits, it will significantly hinder algorithm efficiency when the offline dataset only includes team rewards. This presents a credit assignment problem, aiming to assign credit to the individual agents within the partial observability and emergent behavior. Credit assignment in online MARL is underpinned by well-established methods, but applying these to the offline realm reveals distinct challenges. For instance, COMA (Foerster et al., 2018), an on-policy learning method, cannot be directly applied in offline settings. It is intrinsically designed for continuous policy adjustments based on real-time interactions with the environment, allowing credit assignments that present the ongoing policy of the agent. In offline MARL, agents are reliant on static, pre-collected datasets, often spanning a variety of behavior policies and actions across different time periods. This diversity in data distributions increases the difficulty of assigning credits, given that the nuances of agent contributions are lost in the plethora of policies. Conversely, while off-policy algorithms like SQDDPG (Wang et al., 2020) and SHAQ (Wang et al., 2022a) can be implemented in offline environments, they often fall short in performance. These methods, even if off-policy, were primarily conceived for online scenarios where continuous feedback aids in refining credit assignments. When restricted to static offline data in offline MARL, they miss out on the

essential dynamism and agility needed to accurately understand the intricate interplay within the dataset. Moreover, those methods relying on the Shapley value, while theoretically robust, face inherent challenges in offline settings. Computing the Shapley value demands consideration of every potential agent coalition, which is computationally taxing and necessitates approximations for practicality. In offline MARL, such approximations can lead to imprecise credit assignments due to a loss in precision, potential data inconsistencies from the static nature of past interactions, and scalability issues, especially when numerous agents operate in intricate environments.

In this paper, we propose a new framework, namely **M**ulti-**A**gent **C**ausal **C**redit **A**ssignment (**MACCA**), to address credit assignment in an offline MARL setting. MACCA equates the importance of the credit assignment and how the agent makes the contribution by causal modeling. MACCA first models the generation of individual rewards and team reward from the causal perspective, and construct a graphical representation (as shown in Figure 1) over the involved environment variables, including all the dimensions of states and actions of all agents, the individual rewards and the team rewards. Our method treats team reward as the causal effect of all the individual rewards and provides a way to recover the underlying parametric model, supported by the theoretical evidence of identifiability. In this way, MACCA offers the ability to distinguish the credit of each agent and gain insights into how their states and actions contribute to the individual rewards and further to

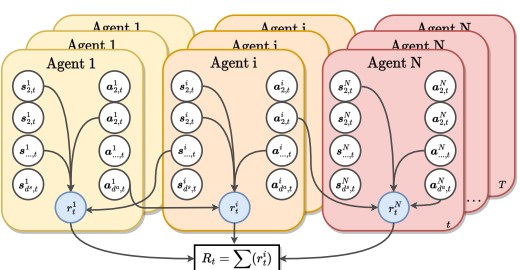

Figure 1: The graphic representation of the causal structure within the MACCA framework. The nodes and edges represent the causal relationships among various environmental variables, i.e., different dimensions of these variables for each agent within the team reward Multi-agent MDP context. These dimensions include the different dimensions of the state $s^i_{\cdots,t}$, action $a^i_{\cdots,t}$, individual reward $r^i_t$ for agent $i$, and the team reward $R_t$. The individual reward $r^i_t$ (shown with blue filling) is unobservable, and the aggregation of $r^i_t$ equals $R_t$.

the team reward. This is achieved through a learned parameterized generative model that decomposes the team reward into individual rewards. The causal structure within the generative process further enhances our understanding by providing insights into the specific contributions of each agent. With the support of theoretical identifiability, we identify the unknown causal structure and individual reward function in such a causal generative process. Additionally, our method offers a clear explanation for actions and states leading to individual rewards, promoting policy optimization and invariance. This clarity enhances agent behavior comprehension and aids in refining policies. The inherent modularity of MACCA ensures its compatibility with a range of policy learning methods, positioning it as a versatile and promising MARL solution for various real-world contexts.

We summarize the main contributions of this paper as follows. First, we reformulate team reward decomposition by introducing a Dynamic Bayesian Network (DBN) to describe the causal relationship among states, actions, individual rewards, and team reward. We provide theoretical evidence of identifiability to learn the causal structure and function within the generation of individual rewards and team rewards. Second, our proposed method can recover the parameterized underlying generative process. Lastly, the empirical results on both discrete and continuous action settings, all three environments, demonstrate that MACCA outperforms current state-of-the-art methods in solving the credit assignment problem caused by team rewards.

## 2 RELATED WORK

In this section, we review the close-related topics, *i.e.*, Offline MARL and Multi-agent Credit Assignment and Causal Reinforcement Learning.

**Offline MARL.** Recent research (Pan et al., 2022; Kostrikov et al., 2021; Jiang & Lu, 2021) efforts have delved into offline MARL, identified and addressed some of the issues inherited from offline single-agent RL (Agarwal et al., 2020; Yu et al., 2020). For instance, ICQ (Kostrikov et al., 2021) focuses on the vulnerability of multi-agent systems to extrapolation errors, while MABCQ (Jiang & Lu, 2021) examines the problem of mismatched transition distributions in fully decentralized offline MARL. However, these studies all assume using a global state and evaluate the action of the agents relying on the team rewards. In contrast, our work MACCA deviates by emphasizing

partially observable settings within offline MARL, which represents a more realistic scenario. Other approaches (Meng et al., 2021; Tseng et al., 2022) have a long term progress in online fine-tuning for offline MARL training but have not taken into account the learning slowdown caused by credits of agents to the entire team. For the learning framework, the two most popular recent paradigms are Centralized Training with Decentralized Execution (CTDE) and Independent Learning (IL). Recent research (de Witt et al., 2020; Lyu et al., 2021) shows the benefits of decentralized paradigms, which lead to more robust performance compared to a centralized value function. In this paper, we use IL as the training paradigm, which not only brings better performance but also increases the scalability of our method. In summary, existing methods have overlooked the exploration of state and action spaces within offline multi-agent datasets, whereas our aim is to bridge this gap.

**Multi-agent Credit Assignment** is the study to decompose the team reward to each individual agent in the cooperative multi-agent environments (Chang et al., 2003). Recent work (Sunehag et al., 2017; Foerster et al., 2018; Wang et al., 2020; Rashid et al., 2020; Yang et al., 2020; Li et al., 2021) focus on value function decompose under online MARL manner. For instance, COMA (Foerster et al., 2018) is a representative method that uses a centralized critic to estimate the counterfactual advantage of an agent action, which is an on-policy algorithm. This means it requires the corresponding data distribution and samples consistent with the current policy for updates. However, in an offline setting, agents are limited to previously collected data and can't interact with the environment. This data, often from varying behavioral policies, might not align with the current policy. Therefore, the COMA cannot be directly extended to the offline setting without changing its on-policy features (Levine et al., 2020). In online off-policy settings, state-of-the-art credit assignment algorithms such as SHAQ (Wang et al., 2022a) and SQDDPG (Wang et al., 2020) utilize an agent's approximate Shapley value for credit assignment. In our experiments section, we conduct a comparative analysis with these methods, and the results for MACCA demonstrate superior performance. Note that we focus on explicitly decomposing the team reward into individual rewards in an offline setting under the casual structure we learned, and these decomposed rewards will be used to reconstruct the offline prioritized dataset and further the policy learning phase.

**Causal Reinforcement Learning.** Plenty of work explores solving diverse RL problems with causal structure. Most conduct research on the transfer ability of RL agents. For instance, Huang et al. (2021) learns factored representation and an individual change factor for different domains, and Feng et al. (2022) extends it to cope with non-stationary changes. More recently, Wang et al. (2022b); Pitis et al. (2022) remove unnecessary dependencies between states and actions variables in the causal dynamics model to improve the generalizing capability in the unseen state, Hu et al. (2023) using causal structure to discover the dependencies between actions and terms of the reward function in order to exploit these dependencies in a policy learning procedure that reduces gradient variance. Also, causal modeling is introduced to multi-agent task (Grimbly et al., 2021; Jaques et al., 2019), model-based RL (Zhang & Bareinboim, 2016), imitation learning (Zhang et al., 2020) and so on. However, most of the previous work does not consider the offline manner and check out the contribution of which dimension of joint state and reward to the individual reward. Compared with the previous work, we investigate the causes for the generation of individual rewards from team rewards in order to help the decentralized policy learning.

## 3 PRELIMINARIES

In this section, we review the widely-used MARL training framework, the Decentralized Partially Observable Markov Decision Process, and briefly introduce Offline MARL.

**Decentralized Partially Observable Markov Decision Process (Dec-POMDP)** is a widely used model for coordination among multiple agents, and it is defined by a tuple $\mathcal{M} = \langle N, \mathcal{S}, \mathcal{A}, \mathcal{P}, \mathcal{R}, \mathcal{O}, \Omega, \gamma \rangle$. In this tuple, $N$ represents the number of agents, $\mathcal{S}$ and $\mathcal{A}$ denote the state and action spaces, respectively. The state transition function $\mathcal{P} : \mathcal{S} \times \mathcal{A} \rightarrow [0, 1]$ specifies the probability of transitioning to a new state given the current state and action. Each agent receives the team reward $R_t$ at time step $t$ based on the team reward function $\mathcal{R} : \mathcal{S} \times \mathcal{A} \rightarrow \mathbb{R}$ and an individual observation $o^i$ from the observation function $\mathcal{O}(s, i) : \mathcal{S} \times \mathcal{A} \rightarrow \Omega$, where $\Omega$ denotes the joint observation space. The objective for each agent is to find an optimal policy $\pi^*$ that maximizes the team discounted return, which is denoted as $\pi^* = \arg\max_\pi \mathbb{E}[\sum_{t=0}^{\infty} \sum_i^N \gamma^t \mathcal{R}(\boldsymbol{s}_t, \boldsymbol{a}_t)]$, where $\gamma$ represents the discount factor. The Dec-POMDP model is flexible and can be used in a wide range of multi-agent scenarios, making it a popular choice for coordination among multiple agents.

**Offline MARL.** Under offline setting, we consider a MARL scenario where agents sample from a fixed dataset $\mathcal{D} = \{s_t^i, o_t^i, a_t^i, R_t, s_t^{i'}, o_t^{i'}\}$. This dataset is generated from the behavior policy $\pi_b$ without any interaction with the environments, meaning that the dataset is pre-collected offline. Here, $s_t^i$, $o_t^i$ and $a_t^i$ represent the state, observation and action of agent $i$ at time $t$, while $R_t$ is the team reward received at time $t$, and $s_t^{i'}$, $o_t^{i'}$ represents the next state and observation of agent $i$.

## 4 OFFLINE MARL WITH CAUSAL CREDIT ASSIGNMENT

Credit assignment plays a crucial role in facilitating the effective learning of policies in offline cooperative scenarios. In this section, we begin with presenting the underlying generative process within the offline MARL scenario, which serves as the foundation of our methods. Then, we show how to recover the underlying generative process and perform policy learning with the assigned individual rewards.

In our method as shown in figure.2, there are two main components, including causal model $\psi_{\mathrm{m}}$ and policy model $\psi_\pi$. The overall objective contains two parts, $L_{\mathrm{m}}$ for model estimation and $J_\pi$ for offline policy learning. Therefore, we minimize the following loss term,

$$L_{\mathrm{MACCA}} = L_{\mathrm{m}} + J_\pi, \tag{1}$$

where $J_\pi$ depends on the applied offline RL algorithms ($J_\pi^{\mathrm{CQR}}$, $J_\pi^{\mathrm{OMAR}}$ or $J_\pi^{\mathrm{ICQ}}$ in this paper.)

### 4.1 UNDERLYING GENERATIVE PROCESS IN MARL

As a foundation of our method, we introduce a Dynamic Bayesian Network (DBN) (Murphy, 2002) to characterize the underlying generative process, leading to a natural interpretation of the explicit contribution of each dimension of state and action towards the individual rewards.

We denote the $\mathcal{G}$ as the DBN to represent the causal structure between the states, actions, individual rewards, and team reward as shown in Figure 1, which is constructed over a finite number of random variables as $(s_{1,t}^i, \cdots, s_{d_s^i, t}^i, a_{1,t}^i, \cdots, a_{d_a^i, t}^i, r_t^i, R_t)_{i,t=1}^{N,T}$, where the $d_s^i$ and $d_a^i$ correspond to the dimensions of the state and action of agent $i$ respectively. $R_t$ is the observed team reward at time step $t$. $r_t^i$ is the unobserved individual reward at time step $t$. $T$ is the maximum episode length of the environment. Then, the underlying generative process is denoted as,

$$\begin{cases} r_t^i = f(\boldsymbol{c}^{i,\boldsymbol{s}\to r} \odot \boldsymbol{s}_t, \boldsymbol{c}^{i,\boldsymbol{a}\to r} \odot \boldsymbol{a}_t, i, \epsilon_{i,t}) \\ R_t = \sum(r_t^1, \cdots r_t^N) \end{cases} \tag{2}$$

where, the $\boldsymbol{s}_t = \{s_{1,t}^1, ..., s_{d_s^1, t}^1, ..., s_{1,t}^N ..., s_{d_s^N, t}^N\}$ and $\boldsymbol{a}_t = \{a_{1,t}^1, ..., a_{d_a^1, t}^1, ..., a_{1,t}^N ..., a_{d_a^N, t}^N\}$ is the joint state and action of all agents at time step $t$. Define $D_{\boldsymbol{s}}$ and $D_{\boldsymbol{a}}$ as the numbers of dimensions of joint state and joint action, where $D_{\boldsymbol{s}} = \sum_{i=1}^N d_{\boldsymbol{s}}^i$ and $D_{\boldsymbol{a}} = \sum_{i=1}^N d_{\boldsymbol{a}}^i$. The $\odot$ is the element-wise product, the $f$ is the unknown non-linear individual reward function, and the $\epsilon_{r,i,t}$ is the i.i.d noise. The masks $\boldsymbol{c}^{i,\boldsymbol{s}\to r} \in \{0,1\}^{D_{\boldsymbol{s}}}$ and $\boldsymbol{c}^{i,\boldsymbol{a}\to r} \in \{0,1\}^{D_{\boldsymbol{a}}}$ are vectors and can be dynamic or static depending on the specific requirements from learning phase, in which control if a specific dimension of the state $\boldsymbol{s}$ and action $\boldsymbol{a}$ impact the individual reward $r_t^i$, separately. Define $c^{j,\boldsymbol{s}\to r}(k)$ as the $k$-th element in the vector $c^{j,\boldsymbol{s}\to r}$. For instance, if there is an edge from the $k$-th dimension of $\boldsymbol{s}$ to the agent $j$'s individual reward $r_t^j$ in $\mathcal{G}$, then the $c^{j,\boldsymbol{s}\to r}(k)$ is 1.

**Proposition 1** (Identifiability of Causal Structure and Individual Reward Function). *Suppose the joint state $\boldsymbol{s}_t$, joint action $\boldsymbol{a}_t$, team reward $R_t$ are observable while the individual $r_t^i$ for each agent are unobserved, and they are from the Dec-POMDP, as described in Eq. 2. Then under the Markov condition and faithfulness assumption (refer to Appendix A), given the current time step's team reward $R_t$, all the masks $\boldsymbol{c}^{i,\boldsymbol{s}\to r}$, $\boldsymbol{c}^{i,\boldsymbol{a}\to r}$, as well as the function $f$ are identifiable.*

The proposition 1 demonstrates that we can identify causal representations from the joint action and state, which serve as the causal parents of the individual reward function we want to fit. This allows us to determine which agent should be responsible for which dimension and thus generate the corresponding individual reward function for each agent. The objective for each agent changes to maximize the sum of individual rewards over an infinite horizon. The proof is in Appendix B.

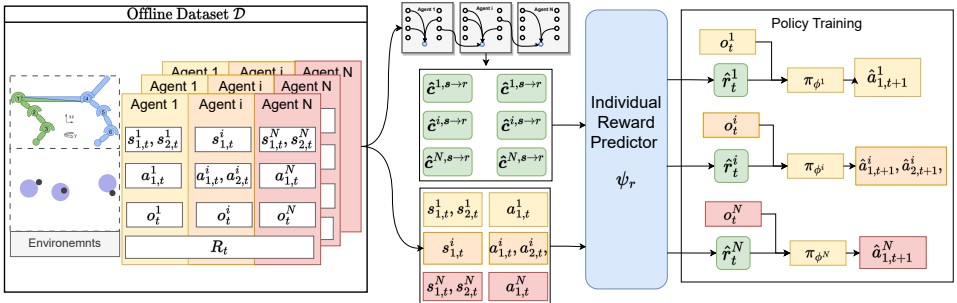

Figure 2: This figure illustrates the workflow of the proposed MACCA structure. The offline data generation process begins on the left side, where data is recorded from the environment. MACCA then constructs a causal model consisting of a DBN represented in grey and an individual reward predictor depicted in blue. The DBN is used to sample scales from each agent, denoted as $c^{i,\cdot\to\cdot}$ and highlighted in green. Meanwhile, the individual reward predictor takes the joint state, action, and these masks as input to generate the individual reward estimate $\hat{r}_t^i$. During the policy learning phase, each agent utilizes their observation and individual reward estimate as inputs, which are then passed through their respective policy network to generate the next-state actions.

## 4.2 CAUSAL MODEL LEARNING

In this section, we delve into identifying the unknown causal structure and reward function within the graph $\mathcal{G}$. This is achieved using the causal structure predictor $\psi_g$, and the individual reward predictor $\psi_r$. The set $\psi_g = \{\psi_g^{s\to r}, \psi_g^{a\to r}\}$ is to learn the causal structure. Specifically, $\psi_g^{s\to r}$ and $\psi_g^{i,a\to r}$ are employed to predict the presence of edges of the mask $c^{i,s\to r} \in \{0,1\}^{d^s}$ and $c^{i,a\to r} \in \{0,1\}^{d^a}$, respectively, as described in Eq. 2. Where,

$$\hat{c}_t^{i,s\to r} = \psi_g^{s\to r}(s_t, a_t, i), \hat{c}_t^{i,a\to r} = \psi_g^{a\to r}(s_t, a_t, i). \tag{3}$$

Here, $\hat{c}_t^{i,s\to r}$ and $\hat{c}_t^{i,a\to r}$ are the predicted masks for agent $i$ at timestep $t$, consider the inherent complexity of the multi-agent scenario, such as the high dimensionality and the dynamic nature of the causal relationships that can evolve over time, we adopt $\psi_g^{s\to r}$ and $\psi_g^{a\to r}$ to generate mask estimation at each time step $t$, within the joint state and joint action and agent id as the input. This dynamic mask adaptation facilitates more accurate causal modeling. To further validate this estimation, we have conducted ablation experiments at section 5.3.

$\psi_r$ is used for approximating the function $f$, and is constructed by stacked fully-connection layers. To recover the underlying generative process, i.e., to optimize $\psi_r$, we minimize the following objective,

$$L_{\mathrm{m}} = \mathbb{E}_{s_t,a_t,R_t\sim\mathcal{D}}[R_t - \sum_{i=1}^N \psi_r(\hat{c}_t^{i,s\to r}, \hat{c}_t^{i,a\to r}, s_t, a_t, i)]^2 + L_{\mathrm{reg}}. \tag{4}$$

The $L_{\mathrm{reg}}$ serves as an L1 regularization, akin to the purpose delineated in Zhang & Spirtes (2011). Its primary objective is to clear redundant features during training, reduce the number of features that a given depends on, and use the coefficients of other features completely set to zero, which fosters model interpretability and mitigates the risk of overfitting. And it defines as,

$$L_{\mathrm{reg}} = \lambda_1 \sum_{i=1}^N \|\hat{c}_t^{i,s\to r}\|_1 + \lambda_2 \sum_{i=1}^N \|\hat{c}_t^{i,a\to r}\|_1. \tag{5}$$

where, $\lambda_{(\cdot)}$ are hyper-parameters. For more details, please refer to Appendix D.

## 4.3 POLICY LEARNING WITH ASSIGNED INDIVIDUAL REWARDS.

For policy learning, we use the redistributed individual rewards $\tilde{r}_t^i$ to replace the observed team reward $R_t$. Then, we carry out the policy optimizing over the offline dataset $\mathcal{D}$.

**Individual Rewards Assignment.** We first assign individual rewards for each agent's state-action-id tuple $\langle s_t, a_t, i \rangle$ in the samples used for policy learning. During such an inference phase of individual rewards predictor, we first utilize a hyper-parameter, $h$, as a threshold to determine the existence of the inference phase. Then the values of $\hat{c}_t^{i,s\to r}$ and $\hat{c}_t^{i,a\to r}$ are set to be zero while their L1-norms are less than $h$. Then, we assign the individual reward for each agent as,

$$\hat{r}_t^i = \psi_r(s_t, a_t, \hat{c}_t^{i,s\to r}, \hat{c}_t^{i,a\to r}, i). \tag{6}$$

Table 1: Average Normalized Score of MPE and MA-MuJoCo task with Team Reward

| | OMAR | I-CQL | MA-ICQ | MACCA-CQL | MACCA-OMAR | MACCA-ICQ |
|---|---|---|---|---|---|---|
| **Exp(CN)** | $44.7 \pm 46.6$ | $33.6 \pm 22.9$ | $45.0 \pm 23.1$ | $85.4 \pm 8.1$ | $\mathbf{111.7 \pm 4.3}$ | $90.4 \pm 5.1$ |
| **Exp(PP)** | $99.9 \pm 14.2$ | $63.4 \pm 38.6$ | $87.0 \pm 12.3$ | $94.9 \pm 27.9$ | $111.0 \pm 21.5$ | $\mathbf{114.4 \pm 25.1}$ |
| **Exp(WORLD)** | $98.7 \pm 18.7$ | $54.4 \pm 17.3$ | $43.2 \pm 15.7$ | $89.3 \pm 14.8$ | $\mathbf{107.4 \pm 11.0}$ | $93.2 \pm 12.0$ |
| **Exp(MA-MuJoCo)** | $110.6 \pm 5.5$ | $66.4 \pm 36.0$ | $77.3 \pm 29.4$ | $81.4 \pm 24.9$ | $\mathbf{113.2 \pm 4.9}$ | $87.4 \pm 9.0$ |
| **Med(CN)** | $49.6 \pm 14.9$ | $19.7 \pm 8.7$ | $30.8 \pm 7.3$ | $45.0 \pm 8.8$ | $67.9 \pm 16.9$ | $\mathbf{70.3 \pm 10.4}$ |
| **Med(PP)** | $57.4 \pm 13.9$ | $50.0 \pm 15.6$ | $59.4 \pm 11.1$ | $61.1 \pm 27.1$ | $\mathbf{87.1 \pm 12.2}$ | $77.4 \pm 10.5$ |
| **Med(WORLD)** | $33.4 \pm 12.8$ | $25.7 \pm 21.3$ | $35.6 \pm 6.0$ | $54.7 \pm 11.0$ | $\mathbf{63.6 \pm 8.7}$ | $55.1 \pm 3.5$ |
| **Med(MA-MuJoCo)** | $64.2 \pm 9.8$ | $48.6 \pm 21.1$ | $55.7 \pm 10.0$ | $50.3 \pm 17.9$ | $\mathbf{66.9 \pm 10.5}$ | $60.0 \pm 11.1$ |
| **Med-R(CN)** | $26.8 \pm 15.2$ | $10.8 \pm 7.7$ | $22.4 \pm 9.3$ | $15.9 \pm 11.2$ | $\mathbf{33.2 \pm 12.6}$ | $28.6 \pm 5.6$ |
| **Med-R(PP)** | $56.3 \pm 16.6$ | $18.3 \pm 9.5$ | $44.2 \pm 4.5$ | $32.5 \pm 15.1$ | $\mathbf{69.0 \pm 19.3}$ | $64.3 \pm 7.8$ |
| **Med-R(WORLD)** | $28.9 \pm 17.2$ | $4.5 \pm 10.1$ | $10.7 \pm 2.8$ | $34.8 \pm 16.7$ | $\mathbf{50.9 \pm 14.2}$ | $39.9 \pm 13.4$ |
| **Med-R(MA-MuJoCo)** | $48.9 \pm 10.4$ | $33.3 \pm 16.1$ | $30.8 \pm 19.2$ | $35.8 \pm 15.4$ | $\mathbf{51.9 \pm 4.5}$ | $47.2 \pm 3.0$ |
| **Rand(CN)** | $22.9 \pm 10.4$ | $12.4 \pm 9.1$ | $6.0 \pm 3.1$ | $22.2 \pm 4.6$ | $\mathbf{32.8 \pm 9.5}$ | $28.13 \pm 4.6$ |
| **Rand(PP)** | $12.0 \pm 5.2$ | $5.5 \pm 2.8$ | $15.6 \pm 3.4$ | $14.7 \pm 6.7$ | $20.9 \pm 8.3$ | $\mathbf{30.3 \pm 5.4}$ |
| **Rand(WORLD)** | $6.2 \pm 6.7$ | $0.1 \pm 4.5$ | $0.6 \pm 2.4$ | $8.7 \pm 3.3$ | $\mathbf{15.8 \pm 6.1}$ | $10.1 \pm 6.6$ |
| **Rand(MA-MuJoCo)** | $7.6 \pm 0.6$ | $6.4 \pm 0.2$ | $3.4 \pm 0.2$ | $10.4 \pm 0.9$ | $\mathbf{20.2 \pm 2.7}$ | $13.4 \pm 3.5$ |

**Offline Policy Learning.** The process of individual reward assignment is flexible and is able to be inserted into any policy training algorithm. We now describe three practical offline MARL methods, MACCA-CQL, MACCA-OMAR and MACCA-ICQ. In all those methods, they use Q-Value to guide policy learning, for each agent who estimates the $Q^i(o^i, a^i) = E_\pi[\sum_{t=0}^{\infty} \gamma^t R_t]$ with the Bellman backup operator, we then replace the team reward by learned individual reward $\hat{r}_t^i$ as $\hat{Q}^i(o^i, a^i) = E_\pi[\sum_{t=0}^{\infty} \gamma^t \hat{r}_t^i]$, then in the policy improvement step, MACCA-CQL trains actors by minimizing:

$$J_\pi^{\text{CQL}} = \mathbb{E}_\mathcal{D}[(\hat{Q}^i(o^i, a^i) - y^i)^2] + \alpha \mathbb{E}_\mathcal{D}[\log \sum_{a^i} \exp(\hat{Q}^i(o^i, a^i)) - \mathbb{E}_{a^i \sim \hat{\pi}_\beta^i}[\hat{Q}^i(o^i, a^i)]], \quad (7)$$

where, $y^i = \hat{r}_t^i + \gamma \min_{k=1,2} \bar{Q}^{i,k}(o^{i'}, \bar{\pi}^i(o^{i'}))$ from (Fujimoto et al., 2018) to minimize the temporal difference error, $\bar{Q}^i$ represents the target $\hat{Q}$ for the agent $i$, $\alpha$ is the regularization coefficient, $\hat{\pi}_{\beta^i}$ is the empirical behavior policy of agent $i$ in the dataset. Similarly, MACCA-OMAR updates actors by minimizing:

$$J_\pi^{\text{OMAR}} = -\mathbb{E}_\mathcal{D}[(1 - \tau)\hat{Q}^i(o^i, \pi^i(o^i)) - \tau(\pi^i(o^i) - \hat{a}_i)^2], \quad (8)$$

where $\hat{a}_i$ is the action provided by the zeroth-order optimizer and $\tau \in [0, 1]$ denotes the coefficient. For the MACCA-ICQ, it updates actors by minimizing:

$$J_\pi^{\text{ICQ}} = \mathbb{E}_\mathcal{D}[L_2^\tau(\hat{r}(s, a) + \gamma \bar{Q}^i(o^{i'}, a^{i'}) - \hat{Q}^i(o^i, a^i))], \quad (9)$$

where $L_2^\tau$ is the squared loss based on expectile regression and the $\gamma$ is the discount factor, which determines the present value of future rewards. As MACCA uses individual reward to replace the team reward, we do not directly decompose value function, unlike the prior offline MARL methods (Foerster et al., 2018; Wang et al., 2020; 2022a), thus we do not require fitting an additional advantage value or Q-value estimator, simplifying our method.

## 5 EXPERIMENTS

Based on the above, our methods include **MACCA-OMAR**, **MACCA-CQL** and **MACCA-ICQ**. For baselines, we compare with both CTDE and independent learning paradigm methods, including **I-CQL** (Yang et al., 2021): conservative Q-learning in independent paradigm, **OMAR** (Pan et al., 2022): based on I-CQL, but learning better coordination actions among agents using zeroth-order optimization, **MA-ICQ** (Kumar et al., 2020): Implicit constraint Q-learning within CTDE paradigm, **SHAQ** (Wang et al., 2022a) and **SQDDPG** (Wang et al., 2020): variants of credit assignment method using Shapley value, which are the SOTA on the online multi-agent RL, **SHAQ-CQL**: In pursuit of a more fair comparison, we integrated CQL with SHAQ, which adopts the architectural framework of SHAQ while using CQL in the estimations of agents' Q-values and the target Q-values, **QMIX-CQL**: conservative Q-learning within CTDE paradigm, following QMIX structure to calculate the $Q^{tot}$ using a mixing layer, which is similar to the MA-ICQ framework. We evaluate those performance in three environments: Multi-agent Particle Environments (MPE) (Lowe et al., 2017), Multi-agent MuJoCo (MA-MuJoCo) (Peng et al., 2021) and StarCraft Micromanagement Challenges (SMAC) (Samvelyan et al., 2019). Through these comparative evaluations, we want to highlight the relative effectiveness and superiority of the MACCA approach. Furthermore, we conduct three ablations to investigate the interpretability and efficiency of our method. For detailed information about the environments, please refer to Appendix C.

Table 2: Averaged win rate of MACCA-based algorithms and baselines in StarCraft II tasks

| Map | Dataset | I-CQL | OMAR | MA-ICQ | MACCA-CQL | MACCA-OMAR | MACCA-ICQ |
|---|---|---|---|---|---|---|---|
| **2s3z** (Easy) | Expert | 0.70±0.09 | 0.86±0.08 | 0.80±0.01 | 0.88±0.07 | **0.99±0.05** | 0.95±0.01 |
| | Medium | 0.20±0.03 | 0.17±0.01 | 0.16±0.07 | 0.27±0.02 | **0.55±0.03** | 0.51±0.03 |
| | Medium-Replay | 0.11±0.07 | 0.35±0.08 | 0.31±0.04 | 0.25±0.03 | 0.53±0.01 | **0.59±0.04** |
| **5m_vs_6m** (Hard) | Expert | 0.02±0.02 | 0.44±0.04 | 0.38±0.05 | 0.63±0.02 | 0.73±0.04 | **0.88±0.01** |
| | Medium | 0.01±0.00 | 0.14±0.02 | 0.11±0.04 | 0.19±0.01 | **0.20±0.04** | 0.15±0.02 |
| | Medium-Replay | 0.12±0.01 | 0.09±0.04 | 0.18±0.04 | 0.15±0.02 | 0.14±0.01 | **0.28±0.01** |
| **6h_vs_8z** (Super Hard) | Expert | 0.00±0.00 | 0.18±0.08 | 0.04±0.01 | 0.59±0.01 | **0.75±0.07** | 0.60±0.03 |
| | Medium | 0.01±0.01 | 0.12±0.06 | 0.01±0.01 | 0.17±0.00 | 0.20±0.02 | **0.22±0.04** |
| | Medium-Replay | 0.03±0.02 | 0.01±0.01 | 0.07±0.04 | 0.14±0.02 | 0.22±0.01 | **0.25±0.05** |
| **MMM2** (Super Hard) | Expert | 0.08±0.03 | 0.10±0.01 | 0.11±0.01 | 0.60±0.01 | 0.69±0.01 | **0.71±0.03** |
| | Medium | 0.02±0.01 | 0.12±0.02 | 0.08±0.04 | 0.25±0.07 | 0.50±0.06 | **0.59±0.04** |

## 5.1 GENERAL IMPLEMENTATION

**Offline Dataset.** Following the approach outlined in (Fu et al., 2020; Pan et al., 2022), we classify the offline datasets in all environments into four types: Random, generated by random initialization. Medium-Reply, collected from the replay buffer until the policy reaches medium performance. Medium and Expert, collected from partially trained to moderately performing policies and fully trained policies, respectively. The difference between our setup and (Pan et al., 2022) is that we hide individual rewards during training and store the sum of these individual rewards in the dataset as the team reward. By creating these different datasets, we aim to explore how different data qualities affect algorithms. For MPE and MA-MuJoCo, we adopt the normalized score as a metric to assess performance. The normalized score is calculated by $100 \times (S - S_{random})/(S_{expert} - S_{random})$ following the (Fu et al., 2020), where the $S, S_{random}, S_{expert}$ are the evaluation return from the current policy, random set behavior policy, expert set behavior policy respectively.

## 5.2 MAIN RESULTS

**Multi-agent Particle Environment (MPE).** We evaluate our method in three distinct environments: Cooperative Navigation (**CN**), Prey-and-Predator (**PP**), and Simple-World (**WORLD**). In the CN environment, three agents aim to reach targets. Observations include position, velocity, and displacements to targets and other agents. Actions are continuous in x and y. Rewards are based on distance to targets, with collision penalties. In the PP environment, three predators chase a random prey. Their state includes position, velocity, and relative displacements. Rewards are based on distance to the prey, with bonuses for captures. The WORLD environment has four allies chasing two faster adversaries. As depicted in Table 1, It can be seen that in all maps and different datasets of MPE, MACCA-based shows better performance than the current state-of-the-art technology. And comparing them with their backbone algorithms respectively, they have improved.

**Multi-agent MuJoCo (MA-MuJoCo).** MA-MuJoCo is a widely-used environment for complex continuous control multi-agent. We consider among them the Half-Cheetah task. In this task, two agents control different parts of the robot joints, and these agents need to cooperate to make the robot move forward by coordinating their movements. The average normalized score for this task across 3 seeds over 1 million time steps, as shown in Table 1. Compared to the baseline, the MACCA-based method outperforms the baseline by approximating expert-level behavior. Consistent competitive performance on Medium, Medium Replay, and random datasets demonstrates their adaptability to different levels of performance and data quality. This success can be attributed to utilizing individual rewards and providing detailed feedback on each agent's actions. By leveraging the task's inherent structure and each agent's specific credit, MACCA achieved higher overall performance levels.

**StarCraft Micromanagement Challenges (SMAC).** In order to show the performance in the scale scene, we specially selected maps with a large number of agents. To illustrate, the map $2s3z$ needs to control 5 agents, including 2 Stalkers and 3 Zealots, the map 6h_vs_8z needs to control 6 Hydralisks against 8 Zealots, and map MMM2 have 1 Medivac, 2 Marauders and 7 Marines. All experiments will run 3 random seeds and the win rate was recorded, and the corresponding standard was calculated. Table 2 shows the result. For most of the tasks, the MACCA-based method shows state-of-the-art performance compared to their baseline algorithms.

Also, we considered testing off-policy algorithms in the offline setting. To this end, we introduced several baselines in SMAC for comparison with MACCA, as shown in Table 3. The table above shows the results of the added baselines compared to SMAC tasks. It becomes evident that the online off-policy credit assignment algorithms, when extended directly to the offline setting, consistently underperform. Our empirical findings underscore that while SHAQ-CQL indeed exhibits advancements QMIX-CQL, our MACCA-CQL clinches the SOTA performance across all tasks.

## 5.3 ABLATION STUDIES

**The impact of learned causal structure.** We varied the value of $\lambda_1$ in Eq.5 to control the sparsity of the learned causal structure. Table 4 presents the average cumulative reward and the sparsity of the causal structure during the training process in the MPE-CN environment. The sparsity of the causal structure $\hat{c}_t^{i,s\rightarrow r}$, is calculated as $\mathcal{S}_{sr} = \sum_{i=1}^{N} \frac{1}{d_s^i} \sum_{k=1}^{d_s^i} s_k^{i,s\rightarrow r}$, where $s_k^{i,s\rightarrow r}$ represent is the value bigger than the threshold $h$. The results indicate that as $\lambda_1$ increases from 0 to 0.5, the causal structure becomes more sparse (sparsity $\mathcal{S}_{sr}$ decreases), resulting in less policy improvement. This can be attributed to the fact that MACCA may not have enough states to predict individual rewards, leading to misguided policy learning accurately. Conversely, setting a relatively low $\lambda_1$ may result in a denser structure that incorporates redundant dimensions, hindering policy learning. Therefore, achieving a reasonable causal structure for the reward function can improve both the convergence speed and the performance of policy training. We also provide the ablation for $\lambda_2$, please refer to Appendix.D.3

**Ground Truth Individual Reward.** In the MPE CN expert dataset, we investigate the influence of ground truth individual rewards on agent policy updates. Two scenarios are compared: agents update policies using ground truth individual rewards (GT), and agents primarily rely on team rewards (without GT). Notably, OMAR with GT directly employs individual rewards

Table 5: Average normalized scores for ground truth individual reward comparison in MPE-CN

|  | OMAR | MACCA-OMAR |
|---|---|---|
| With GT | $114.9 \pm 2.4$ | $113.7 \pm 2.3$ |
| Without GT | $43.7 \pm 46.6$ | $111.7 \pm 4.3$ |

for policy updates, while MACCA-OMAR with GT utilizes individual rewards as a supervisory signal, replacing team rewards in Eq. 4. The results, presented in Table 5, demonstrate that MACCA-OMAR with GT achieves similar performance to OMAR with GT. Although MACCA-OMAR with GT exhibits slightly slower convergence and performance due to the learning of unbiased causal structures and individual reward functions, it overcomes this drawback by incorporating individual rewards as supervisory signals, mitigating the bias associated with relying solely on team rewards. More Importantly, MACCA-OMAR effectively addresses the challenge of exclusive team reward reliance by attaining a more comprehensive understanding of individual credits through the causal structure and individual reward function. These findings demonstrate that while MACCA-OMAR's performance is slightly lower than that of OMAR under GT, it offers the advantage of mitigating the bias caused by relying solely on team rewards.

**Different Causal Graph Setting.** To investigate how various causal graph settings affect the algorithm's performance, we performed an ablation study using the expert data set from the SMAC 5m_vs_6m map. According to the Table 6, here the **FCG** stands for using fully connected causal graph as the mask ($\hat{c}_t^{i,\cdot\rightarrow r}(k)=1$), the **FG** is to learn a fixed graph without time variants ($\hat{c}^{i,\cdot\rightarrow r}$, without timestep), **DCG** is to learn a dynamic

Table 6: Average win rate in SMAC 5m_vs_6m map, expert dataset.

|  | OMAR |
|---|---|
| Backbone | $0.44 \pm 0.04$ |
| MACCA (FCG) | $0.38 \pm 0.02$ |
| MACCA (FG w. $h$ clipping) | $0.50 \pm 0.01$ |
| MACCA (DG w.o $h$ clipping) | $0.66 \pm 0.01$ |
| MACCA (DG w. $h$ clipping) | $\mathbf{0.73 \pm 0.04}$ |

causal graph ($\hat{c}_t^{i,\cdot\rightarrow r}$). The $h$ clipping means using hyper-parameter threshold $h$ to filter the causal mask. Utilizing a fully connected causal graph (FCG) indicates that all states and actions directly influence the reward, resulting in suboptimal performance. This indicates an inability in the current setting to differentiate the individual contributions of agents to the collective reward. While the performance of learning a mask without time variants (FG) shows a marginal improvement over the baseline, the enhancement remains minimal. This can be attributed to the inherent challenges in directly learning a comprehensive multi-agent causal graph, especially given the intricacies of

Table 3: Compare with online off-policy credit assignment baselines in SMAC

| Map | Dataset | SHAQ | SQDDPG | SHAQ-CQL | QMIX-CQL | ICQL | MACCA-CQL |
|---|---|---|---|---|---|---|---|
| **2s3z** | Expert | 0.10±0.03 | 0.05±0.01 | 0.79±0.03 | 0.73±0.02 | 0.70±0.09 | **0.88±0.07** |
|  | Medium | 0.05±0.03 | 0.07±0.01 | 0.24±0.01 | 0.22±0.03 | 0.20±0.03 | **0.27±0.02** |
| **5m_vs_6m** | Expert | 0.02±0.01 | 0.00±0.00 | 0.10±0.03 | 0.03±0.01 | 0.02±0.02 | **0.63±0.02** |
|  | Medium | 0.00±0.00 | 0.00±0.00 | 0.06±0.01 | 0.01±0.01 | 0.01±0.00 | **0.19±0.01** |
| **6h_vs_8z** | Expert | 0.00±0.00 | 0.00±0.00 | 0.02±0.01 | 0.00±0.00 | 0.00±0.00 | **0.59±0.01** |
|  | Medium | 0.00±0.00 | 0.00±0.00 | 0.04±0.02 | 0.00±0.00 | 0.01±0.01 | **0.17±0.00** |

Table 4: The mean and the standard variance of average normalized score, sparsity rate $\mathcal{S}_{sr}$ of $\hat{c}_t^{i,s\to r}$ with diverse $\lambda_1$ at different time step $t$ in MPE-CN.

| $\lambda_1$ / $t$ | 1e4 | 3e4 | 5e4 | 1e5 | 2e5 |
|---|---|---|---|---|---|
| 0 | -2.43 ± 8.01(0.98) | -14.87± 7.71(0.90) | -12.356± 5.83(0.81) | 9.842± 18.89(0.77) | 69.04 ± 19.69(0.72) |
| 0.007 | -7.88±5.36(0.94) | **13.26±27.14(0.47)** | **60.18±26.14(0.28)** | **99.78± 19.50(0.15)** | **111.65± 4.28(0.13)** |
| 0.05 | -3.66±12.14(0.90) | 3.93±42.06(0.34) | 10.04± 45.97(0.17) | 23.61± 44.18(0.11) | 75.81± 34.48(0.10) |
| 0.5 | -12.20±3.87(0.87) | -16.19±5.53(0.24) | -8.84± 7.16(0.11) | 16.40± 21.04(0.07) | 59.23± 35.29(0.01) |

the environment. Similarly, the efficacy of a learned causal graph without threshold clipping (i.e., w.o $h$ clipping) is slightly superior to the baseline but doesn't match the performance of DG with $h$ clipping. In real-world implementations and when working with finite datasets, the model often finds it challenging to ensure edge weights converge precisely to zero. Even when sparsity loss and normalization are introduced, threshold clipping remains indispensable. Such an approach aligns with established practices in causal structure discovery with continuous optimization, as evidenced by (Zheng et al., 2018; Ng et al., 2020).

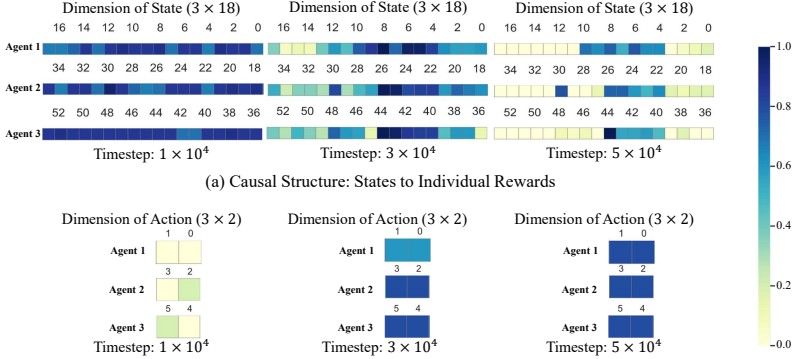

(a) Causal Structure: States to Individual Rewards

(b) Causal Structure: Actions to Individual Rewards

Figure 3: The figure visualizes the causal structure, showing the probability of causal edges from blue (high probability) to yellow (low probability). **(a)** represents the causal structure $\hat{c}_t^{i,s\to r}$ between the state of all agents (18 dimensions for each agent, 54 dimensions for joint state ) and the individual reward (1 dimension for each agent). **(b)** represents the causal structure $\hat{c}_t^{i,a\to r}$ between the action of each agent (2 dimensions for each agent, six dimensions for joint action) and the individual reward (1 dimension for each agent).

**Visualization of Causal Structure.** In Figure D.3, we present visualizations of two significant causal structures within the CN environment of MPE. To facilitate the observation of the causal structure learning process, we initialize S2R as a normalized random number close to 1 and A2R as a normalized random number close to 0. As time progresses, we observe that the causal structure $\hat{c}_t^{i,s\to r}$ transitions its focus from considering all dimensions of the agent state to primarily emphasizing the 4th to 10th dimensions of each agent. By analyzing the state output of the environment, we determine that each agent's state comprises 18 dimensions. Specifically, dimensions 0-4 represent the agent's velocity and position on the x and y axes, dimensions 4-9 capture the distance between the agent and three distinct landmarks on the x and y axes, dimensions 10-13 reflect the distances between the agent and other agents and dimensions 14-17 are related to communication, although not applicable in this experiment and thus considered as irrelevant. Variables 4-9 and 10-13 are intuitively linked to individual rewards, aligning with the convergence direction of MACCA. Regarding the causal structure $\hat{c}_t^{i,a\to r}$, as each agent's actions involve continuous motion without extraneous variables, it converges to relevant states that contribute to individual credits for the team reward. The experimental results demonstrate that MACCA exhibits rapid convergence, facilitating the learning of interpretable causal structures within short time steps. Therefore, our findings support the interpretability of the causal structure and its ability to provide a clear understanding of the relationships between variables.

## 6 CONCLUSION

In conclusion, MACCA emerges as a valuable solution to the credit assignment problem in offline Multi-agent Reinforcement Learning (MARL), providing an interpretable and modular framework for capturing the intricate interactions within multi-agent systems. By leveraging the inherent causal structure of the system, MACCA allows us to disentangle and identify the specific credits of individual agents to team rewards. This enables us to accurately assign credit and update policies accordingly,

leading to enhanced performance compared to different baseline methods. The MACCA framework empowers researchers and practitioners to gain deeper insights into the dynamics of multi-agent systems, facilitating the understanding of the causal factors that drive cooperative behavior and ultimately advancing the capabilities of MARL in a variety of real-world applications.

## 7 REPRODUCIBILITY STATEMENT

To promote transparent and accountable research practices, we have prioritized the reproducibility of our method. All experiments conducted in this study adhere to controlled conditions and well-known environments and datasets, with detailed descriptions of the experimental settings available in Section 5 and Appendix C. The implementation specifics for all the baseline methods and our proposed MACCA are thoroughly outlined in Section 4 and Appendix D.

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

## A    MARKOV AND FAITHFULNESS ASSUMPTIONS

A directed acyclic graph (DAG), $\mathcal{G} = (\boldsymbol{V}, \boldsymbol{E})$, can be deployed to represent a graphical criterion carrying out a set of conditions on the paths, where $\boldsymbol{V}$ and $\boldsymbol{E}$ denote the set of nodes and the set of directed edges, separately.

**Definition 1.** *(d-separation (Pearl, 2000)). A set of nodes $\boldsymbol{Z} \subseteq \boldsymbol{V}$ blocks the path $p$ if and only if (1) $p$ contains a chain $i \rightarrow m \rightarrow j$ or a fork $i \leftarrow m \rightarrow j$ such that the middle node $m$ is in $\boldsymbol{Z}$, or (2) $p$ contains a collider $i \rightarrow m \leftarrow j$ such that the middle node $m$ is not in $\boldsymbol{Z}$ and such that no descendant of $m$ is in $\boldsymbol{Z}$. Let $\boldsymbol{X}$, $\boldsymbol{Y}$ and $\boldsymbol{Z}$ be disjunct sets of nodes. If and only if the set $\boldsymbol{Z}$ blocks all paths from one node in $\boldsymbol{X}$ to one node in $\boldsymbol{Y}$, $\boldsymbol{Z}$ is considered to d-separate $\boldsymbol{X}$ from $\boldsymbol{Y}$, denoting as $(\boldsymbol{X} \perp_d \boldsymbol{Y} \mid \boldsymbol{Z})$.*

**Definition 2.** *(Global Markov Condition (Spirtes et al., 2000; Pearl, 2000)). If, for any partition $(\boldsymbol{X}, \boldsymbol{Y}, \boldsymbol{Z})$, $\boldsymbol{X}$ is d-separated from $\boldsymbol{Y}$ given $\boldsymbol{Z}$, i.e. $\boldsymbol{X} \perp_d \boldsymbol{Y} \mid \boldsymbol{Z}$. Then the distribution $P$ over $\boldsymbol{V}$ satisfies the global Markov condition on graph $G$, and can be factorizes as, $P(\boldsymbol{X}, \boldsymbol{Y} \mid \boldsymbol{Z}) = P(\boldsymbol{X} \mid \boldsymbol{Z})P(\boldsymbol{Y} \mid \boldsymbol{Z})$. That is, $\boldsymbol{X}$ is conditionally independent of $\boldsymbol{Y}$ given $\boldsymbol{Z}$, writing as $\boldsymbol{X} \perp\!\!\!\perp \boldsymbol{Y} \mid \boldsymbol{Z}$.*

**Definition 3.** *(Faithfulness Assumption (Spirtes et al., 2000; Pearl, 2000)). The variables, which are not entailed by the Markov Condition, are not independent of each other.*

*Under the above assumptions, we can apply d-separation as a criterion to understand the conditional independencies from a given DAG $G$. That is, for any disjoint subset of nodes $\boldsymbol{X}, \boldsymbol{Y}, \boldsymbol{Z} \subseteq \boldsymbol{V}$, $(\boldsymbol{X} \perp\!\!\!\perp \boldsymbol{Y} \mid \boldsymbol{Z})$ and $\boldsymbol{X} \perp_d \boldsymbol{Y} \mid \boldsymbol{Z}$ are the necessary and sufficient condition of each other.*

## B    PROOF OF IDENTIFIABILITY

**Proposition 1** (Individual Reward Function Identifiability). *Suppose the joint state $\boldsymbol{s}_t$, joint action $\boldsymbol{a}_t$, team reward $R_t$ are observable while the individual $r_t^i$ for each agent are unobserved, and they are from the Dec-POMDP, as described in Eq. 2. Then, under the Markov condition and faithfulness assumption, given the current time step's team reward $R_t$, all the masks $\boldsymbol{c}^{\boldsymbol{s} \rightarrow r, i}$, $\boldsymbol{c}^{\boldsymbol{a} \rightarrow r, i}$, as well as the function $f$ are identifiable.*

**Assumption**    We assume that, $\epsilon_{i,t}$ in Eq. 2 are i.i.d additive noise. From the weight-space view of Gaussian Process (Williams & Rasmussen, 2006) and equation.6, equivalently, the causal models for $r_t^i$ can be represented as follows,

$$r_t^i = f(\boldsymbol{c}_t^{i, \boldsymbol{s} \rightarrow r} \odot \boldsymbol{s}_t, \boldsymbol{c}_t^{i, \boldsymbol{a} \rightarrow r} \odot \boldsymbol{a}_t, i) + \epsilon_{r,t} = W_f^T \phi_r(\boldsymbol{s}_t, \boldsymbol{a}_t, i) + \epsilon_{i,t} \tag{A1}$$

where $\forall i \in [1, N]$, and $\phi_r$ denote basis function sets.

As $\boldsymbol{s}_t = \{s_{1,t}^1, ..., s_{d_s^1, t}^1, ..., s_{1,t}^N, ..., s_{d_s^N, t}^N\}$ and $\boldsymbol{a}_t = \{a_{1,t}^1, ..., a_{d_a^1, t}^1, ..., a_{1,t}^N, ..., a_{d_a^N, t}^N\}$. We denote the variable set in the system by $\boldsymbol{V} = \{\boldsymbol{V}_0, ..., \boldsymbol{V}_T\}$, where $\boldsymbol{V}_t = \boldsymbol{s}_t \cup \boldsymbol{a}_t \cup R_t$, and the variables form a Bayesian network $\mathcal{G}$. Following AdaRL (Huang et al., 2021), there are possible edges only from $\boldsymbol{s}_{k,t}^i \in \boldsymbol{s}_t$ to $r_t^i$, and from $\boldsymbol{a}_{j,t}^i \in \boldsymbol{a}_t$ to $r_t^i$ in $\mathcal{G}$, where $k, j$ are dimension index in $[1, ..., d_s^N]$ and $[1, ..., d_a^N]$ respectively. In particular, the $r_t^i$ are unobserved, while $R_t = \sum_{i=1}^N r_t^i$ is observed. Thus, there are deterministic edges from each $r_t^i$ to $R_t$.

**Proof**    We aim to prove that, given the team reward $R_t$, and the $\boldsymbol{c}^{i, \boldsymbol{s} \rightarrow r}$, $\boldsymbol{c}^{i, \boldsymbol{a} \rightarrow r}$ and $r_t^i$ are identifiable. Following the above assumption, we can rewrite the Eq.2 to the following,

$$
\begin{aligned}
R_t &= \sum_{i=1}^N r_t^i \\
&= \sum_{i=1}^N \left[ W_f^T \phi_r(\boldsymbol{s}_t, \boldsymbol{a}_t, i) + \epsilon_{i,t} \right] \\
&= W_f^T \sum_{i=1}^N \phi_r(\boldsymbol{s}_t, \boldsymbol{a}_t, i) + \sum_{i=1}^N \epsilon_{i,t}.
\end{aligned}
\tag{A2}
$$

For simplicity, we replace the components in Eq. A2 by,

$$
\begin{aligned}
\Phi_{r,t} &= \sum_{i=1}^{N} \phi_r(\boldsymbol{s}_t, \boldsymbol{a}_t, i), \\
\mathcal{E}_{r,t} &= \sum_{i=1}^{N} \epsilon_{i,t}.
\end{aligned}
\tag{A3}
$$

Consequently, we derive the following equation,

$$
R_t = W_f{}^T \Phi_{r,t}(X_t) + \mathcal{E}_{r,t},
\tag{A4}
$$

where $X_t := [\boldsymbol{s}_t, \boldsymbol{a}_t, i]_{i=1}^{N}$ representing the concatenation of the covariates $\boldsymbol{s}_t$ , $\boldsymbol{a}_t$ and $i$, from $i = 1$ to $N$.

Then we can obtain a closed-form solution of $W_f{}^T$ in Eq. A4 by modelling the dependencies between the covariates $X_t$ and response variables $R_t$. One classical approach to finding such a solution involves minimizing the quadratic cost and incorporating a weight-decay regularizer to prevent overfitting. Specifically, we define the cost function as,

$$
C(W_f) = \frac{1}{2} \sum_{X_t, R_t \sim \mathcal{D}} (R_t - W_f{}^T \Phi_{r,t}(X_t))^2 + \frac{1}{2}\lambda \|W_f\|^2.
\tag{A5}
$$

where $X_t$ and long-term returns $R_t$, which are sampled from the offline dataset $\mathcal{D}$. $\lambda$ is the weight-decay regularization parameter. To find the closed-form solution, we differentiate the cost function with respect to $W_f$ and set the derivative to zero:

$$
\frac{\partial C(W_f)}{\partial W_f} \to 0.
\tag{A6}
$$

Solving Eq.A6 will yield the closed-form solution for $W_f$, as

$$
W_f = (\lambda I_d + \Phi_{r,t}\Phi_{r,t}{}^T)^{-1}\Phi_{r,t}R_t = \Phi_{r,t}(\Phi_{r,t}{}^T \Phi_{r,t} + \lambda I_n)^{-1}R_t
\tag{A7}
$$

Therefore, $W_f$, which indicates the causal structure and strength of the edge, can be identified from the observed data. In summary, given team reward $R_t$, the binary masks, $\boldsymbol{c}^{i,\boldsymbol{s}\to r}$, $\boldsymbol{c}^{i,\boldsymbol{a}\to r}$ and individual $r_t^i$ are identifiable.

Considering the Markov condition and faithfulness assumption, we can conclude that for any pair of variables $V_k, V_j \in \boldsymbol{V}$, $V_k$ and $V_j$ are not adjacent in the causal graph $\mathcal{G}$ if and only if they are conditionally independent given some subset of $\{V_l \mid l \neq k, l \neq j\}$. Additionally, since there are no instantaneous causal relationships and the direction of causality can be determined if an edge exists, the binary structural masks $\boldsymbol{c}^{i,\boldsymbol{s}\to r}$ and $\boldsymbol{c}^{i,\boldsymbol{a}\to r}$ defined over the set $\boldsymbol{V}$ are identifiable with conditional independence relationships (Huang et al., 2022). Consequently, the functions $f$ in Equation 2 are also identifiable.

## C  ENVIRONMENTS SETTING

We adopt the open-source implementations for the multi-agent particle environment (Lowe et al., 2017)[1] and the Multi-agent MuJoCo (Peng et al., 2021)[2]. The tasks in the multi-agent particle environments are illustrated in Figures A1(a)-(c), while Figure A1(d) depicts the MA-MuJoCo half-cheetah task. The Cooperative Navigation (CN) task involves 3 agents and 3 landmarks, requiring agents to cooperate in covering the landmarks without collisions. In the Predator-Prey (PP) task, 3 predators collaborate to capture prey that is faster than them. Finally, the WORLD task features 4 slower cooperating agents attempting to catch 2 faster adversaries, with the adversaries aiming to consume food while avoiding capture. The MA-MuJoCo half-cheetah task depicted in Figure A1(d) involves a cooperative scenario where multiple agents coordinate their actions to control a half-cheetah robot. The objective is to achieve a desired gait pattern and maintain stability while navigating the environment.

---

[1]https://github.com/openai/multiagent-particle-envs
[2]https://github.com/schroederdewitt/multiagent_mujoco

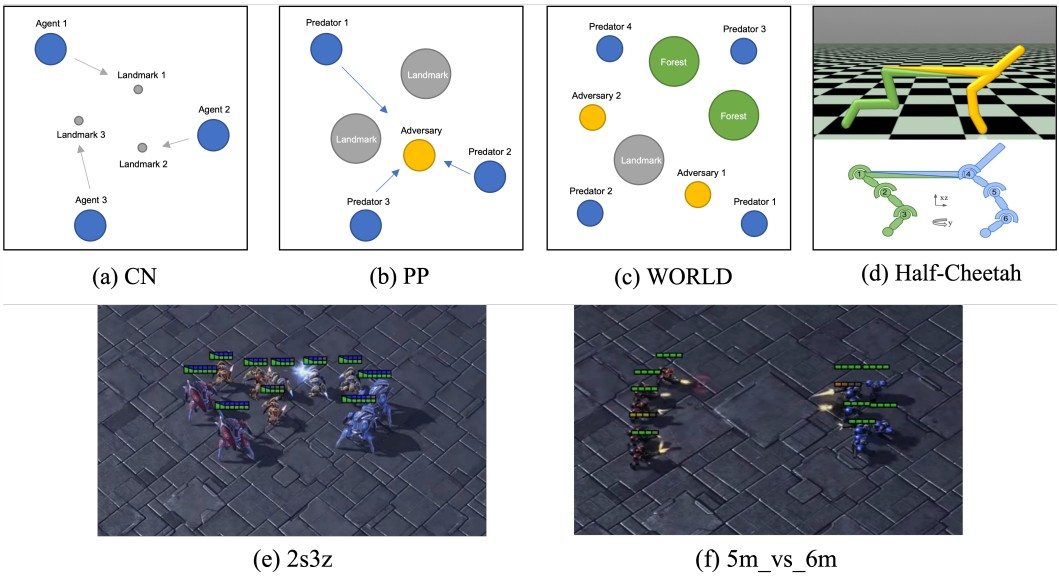

(a) CN  (b) PP  (c) WORLD  (d) Half-Cheetah

(e) 2s3z  (f) 5m_vs_6m

Figure A1: Visualization of different environment in the experiments, **(a)-(c):** Multi-agent Particle Environments (MPE), **(d):** Multi-agent MuJoCo (MA-MuJoCo), **(e)-(f):** StarCraft Micromanagement Challenges (SMAC)

**Datasets.** During training, we utilize the team reward as input, while for evaluation purposes, we compare the performance with the ground truth individual reward. As a result, the expert and random scores for the Cooperative Navigation, Predator-Prey, World, and Half-Cheetah tasks are as follows: Cooperative Navigation - expert: 516.526, random: 160.042; Predator-Prey - expert: 90.637, random: -2.569; World - expert: 34.661, random: -8.734; Half-Cheetah - expert: 3568.8, random: -284.0.

## D  IMPLEMENTATIONS

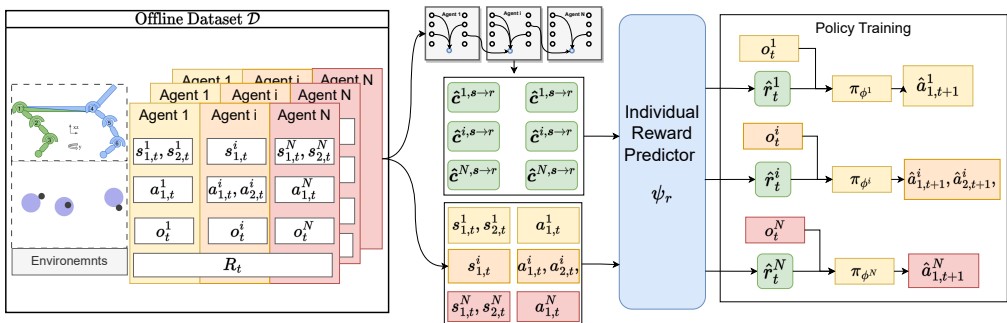

Figure A2:  This figure illustrates the workflow of the proposed MACCA structure. The offline data generation process begins on the left side, where data is recorded from the environment, including actions $a_t^i$, states $s_t^i$ and observations $o_t^i$ of each agent, as well as the team reward $R_t$ provided by the environment. MACCA then constructs a causal model consisting of a DBN represented in grey and an individual reward predictor depicted in blue. The DBN is used to sample scales from each agent, denoted as $c^{i,\cdot\to\cdot}$ and highlighted in green. Meanwhile, the individual reward predictor takes the joint state, action, and these masks as input to generate the individual reward estimate $\hat{r}_t^i$. During the policy learning phase, each agent utilizes their observation and individual reward estimate as inputs, which are then passed through their respective policy network $\phi^i$ to generate the next-state actions.

## D.1 ALGORITHM

---

**Algorithm 1** MACCA: **M**ulti-**A**gent **C**ausal **C**redit **A**ssignment

---
1: **for** training step $t = 1$ to $T$ **do**
2:     Sample trajectories from $\mathcal{D}$, save in minibatch $\mathcal{B}$
3:     **for** agent $i = 1$ to $N$ **do**
4:         Update the team reward $R_t$ to $\hat{r}_t^i$ in $\mathcal{B}$ (Eq. 6)
5:         Optimize $\psi_m$: $\psi_m \leftarrow \psi_m - \alpha\nabla_{\psi_m}L_m$ (Eq. 4)
6:     **end for**
7:     Update policy $\pi$ with minibatch $\mathcal{B}$ (Eq. 7, Eq. 8 or Eq. 9)
8:     Reset $\mathcal{B} \leftarrow \emptyset$
9: **end for**

---

## D.2 MODEL STRUCTURE

The parametric generative model $\psi_m$ used in MACCA consists of two parts: $\psi_g$ and $\psi_r$. The function of $\psi_g$ is to predict the causal structure, which determines the relationships between the environment variables. The role of $\psi_r$ is to generate individual rewards based on the joint state and action information. This prediction is achieved through a network architecture that includes three fully-connected layers with an output size of 256, followed by an output layer with a single output. Each hidden layer is activated using the rectified linear unit (ReLU) activation function.

During the training process, the generative model is optimized to learn the causal structure and generate individual rewards that align with the observed team rewards. The model parameters are updated using Adam, to minimize the discrepancy between the predicted sum of individual rewards and the team rewards. The training process involves iteratively adjusting the parameters to improve the accuracy of the predictions.

For a more detailed overview of the training process, including the specific loss functions and optimization algorithms used, please refer to Fig A2. The figure provides a step-by-step illustration of the training pipeline, helping to visualize the flow of information and the interactions between different components of the generative model.

Table A1: The Common Hyper-parameters.

| hyperparameters | value | hyperparameters | value |
|---|---|---|---|
| steps per update | 100 | optimizer | Adam |
| batch size | 1024 | learning rate | $3 \times 10^{-4}$ |
| hidden layer dim | 64 | $\gamma$ | 0.95 |
| evaluation interval | 1000 | evaluation episodes | 10 |

Table A2: Hyper-parameters for OMAR, CQL and MACCA

| | OMAR $\tau$ | CQL $\alpha$ | MACCA $\lambda_1$ | MACCA $\lambda_2$ | MACCA $r_{lr}$ | MACCA $h$ |
|---|---|---|---|---|---|---|
| Expert | 0.9 | 5.0 | 7e-3 | 7e-3 | 5e-2 | 0.1 |
| Medium | 0.7 | 0.5 | 5e-3 | 5e-3 | 5e-2 | 0.1 |
| Medium-Replay | 0.7 | 1.0 | 5e-3 | 7e-3 | 5e-2 | 0.1 |
| Random | 0.99 | 1.0 | 1e-7 | 1e-3 | 5e-2 | 0.1 |

## D.3 HYPER-PARAMETERS

The neural network used in training is initialized from scratch and optimized using the Adam optimizer with a learning rate of $3 \times 10^{-4}$. The policy learning process involves varying initial learning rates based on the specific algorithm, while the hyperparameters for policy learning, including a discount factor of 0.95, are consistent across all tasks.

The training procedure differs across tasks. For MPE, the training duration ranges from 20,000 to 60,000 iterations, with longer training for behavior policies that perform poorly. The number of steps per update is set to 100. In MA-MuJoCo, training comprises 1 million time steps, and the number of steps per update is reduced to 10.

During each training iteration, trajectories are sampled from the offline data, and the generated individual reward is replaced with the team reward for policy updates. The training of $\psi_{\text{cau}}$ is performed concurrently with $\psi_{\text{rew}}$. Validation is conducted after each epoch, and the average metrics are computed using 5 random seeds for reliable evaluation.

The hyperparameters specific to training MACCA models can be found in Table A2. All experiments were conducted on a high-performance computing (HPC) system featuring 128 Intel Xeon processors running at 2.2 GHz, 5 TB of memory, and an Nvidia A100 PCIE-40G GPU. This computational setup ensures efficient processing and reliable performance throughout the experiments.

# E    ADDITIONAL RESULT

## E.1    ABLATION FOR $\lambda_2$

We have conducted ablation experiments on $\lambda_2$ and show the results in the table.A3

Table A3: The mean and the standard variance of average normalized score, sparsity rate $\mathcal{S}_{ar}$ of $\hat{c}_t^{i,a\rightarrow r}$ with diverse $\lambda_2$ at different time step $t$ in MPE-CN.

| $\lambda_2$ / $t$ | 1e4 | 5e4 | 1e5 | 2e5 |
|---|---|---|---|---|
| 0 | 17.4 ± 15.2(0.98) | 93.1 ± 6.4 (1.0) | 105 ± 3.5 (1.0) | 107.7 ± 10.2 (1.0) |
| 0.007 | 19.9 ± 12.4 (0.8 | **90.2 ± 7.1 (1.0)** | **108.8 ± 4.0 (1.0)** | **111.7 ± 4.3(1.0)** |
| 0.5 | 13.3 ± 11.1 (0.68) | 100.5 ± 14.0 (0.84) | 102.9 ± 16.4 (0.87) | 108.4 ± 6.4 (0.98) |
| 5.0 | 2.3 ± 9.8 (0.0) | -1.3 ± 25.4 (0.34) | 70.4 ± 18.0 (0.62) | 100.1 ± 7.4 (0.75) |

This table shows the mean and the standard variance of the average normalized score with diverse $\lambda_2$ in the MPE-CN task. The value in brackets is the sparsity rate $\mathcal{S}_{ar}$ of $\hat{c}_t^{i,a\rightarrow r}$, whose definition can be found in Section 5.3. For all values of $\lambda_2$, the sparsity rate $\mathcal{S}_{ar}$ consistently begins from zero. Over time, there is a discernible increase in $\mathcal{S}_{ar}$, and the convergence speed slows down with the increase of $\lambda_2$. This pattern intimates that higher $\lambda_2$ values engender a more measured modulation in the causal impact exerted by actions on individual rewards. Furthermore, despite the variation in $\lambda_2$ values, the average normalized scores across different $\lambda_2$ settings eventually converge towards a similar level.

## E.2    MACCA IN ONLINE TEST

To prove the versatility and efficacy of the MACCA framework in various settings, we have recently applied it to the online environment of the SMAC in combination with Independent Q-Learning (IQL). This new integration, referred to as MACCA-IQL, was specifically designed to harness the strengths of MACCA in credit assignment and the robustness of IQL in an online context. Our experiments, which provide a comparative analysis against Multi-Agent Proximal Policy Optimization (MAPPO) in the SMAC 3m and MMM2 environment, demonstrate the effectiveness of this approach.

To elucidate our choice of combining MACCA with an off-policy algorithm like IQL, rather than an on-policy method, it is crucial to consider the distinct advantages offered by off-policy learning in our context. Off-policy algorithms, such as IQL , allow for more efficient utilization of past experiences, enabling learning from a broader range of data and reducing the dependency on current policy performance. This is particularly beneficial in causal modelling in complex multi-agent settings where the diversity of experiences can enhance learning efficiency. Thus, our decision to integrate

MACCA with an off-policy approach was driven by the goal of maximizing learning efficiency and stability in the challenging context of online multi-agent reinforcement learning.

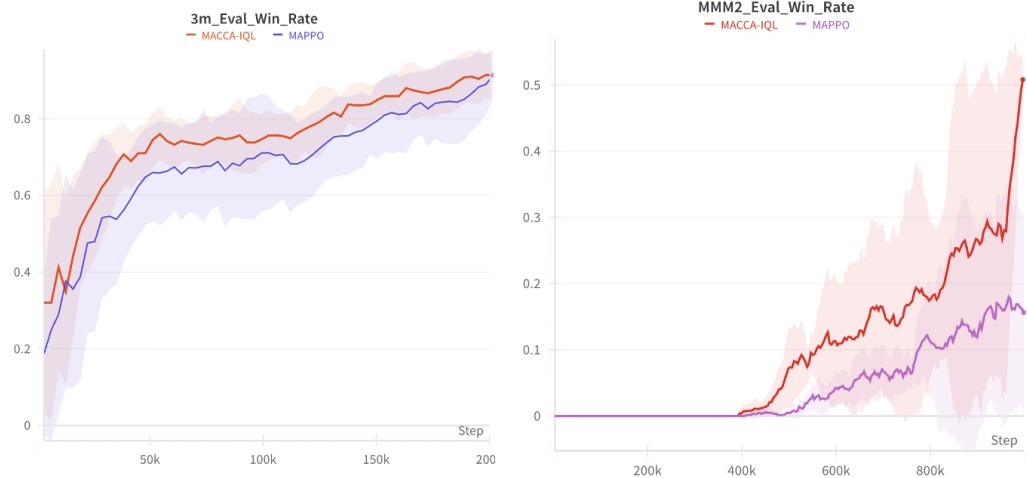

Figure A3: The win rate for MACCA-IQL compared with MAPPO in the SMAC online setting in 3m (easy) and MMM2 (Super-Hard).

According to the figure. D.3. The adaptability and potential of MACCA-IQL in enhancing policy learning in the domain of online setting is demonstrated. It is worth noting that in 3m due to the simple map and the small number of three agents, the improvement is limited, while in MMM2 with more agents, a more obvious improvement is achieved.

