# OpenReview forum: "MACCA: Offline Multi-agent Reinforcement Learning with Causal Credit Assignment"
_ICLR.cc/2024/Conference — Submitted to ICLR 2024_

### Official Review · Reviewer_SGeQ · 2023-10-30

**Soundness:** 2 fair
**Presentation:** 2 fair
**Contribution:** 2 fair
**Rating:** 3
**Confidence:** 4

**Summary:**

MACCA is a reward decomposition mechanism in offline MARL, which follows centralized training and decentralized execution. It factorizes the shared global reward into individual rewards according to the causal relationship among states, actions, and rewards. The individual rewards can be seamlessly optimized by existing offline MARL methods. The experiments are performed on offline datasets of MPE, MA-MuJoCo, and SMAC.

**Strengths:**

+ The paper is well-organized.
+ The experiments are extensive, and MACCA achieves performance gain.
+ The experiment settings and hyper-parameters are detailed.

**Weaknesses:**

There are two main drawbacks in this paper:

The target of credit assignment is not the immediate reward, but the cumulative reward of the whole trajectory, which is affected by the immediate reward and the transition (long-horizon rewards). However, this paper only considers the decomposition of immediate reward, ignoring the long-horizon causal relationship. For example, in the delayed reward setting (SMAC), the agents only receive the reward at the last timestep, the credit assignment of the whole trajectory in MACCA is only related to the state and actions of the last timestep, which is unreasonable.

The states of the environments adopted by this paper are low-dimension vectors. It is hard to capture the causal relationship between image state and reward.

**Questions:**

The proposed reward decomposition mechanism is not specific to offline MARL. The results will be more convincing if MACCA achieves performance gain in the online MARL (SMAC tasks).

---

> ### Author Response · Authors · 2023-11-16
> **Response to Reviewer SGeQ (1/2)**
>
> We appreciate your review and provide point-by-point responses to your questions below.
>
> > **Weakness 1**: The target of credit assignment is not the immediate reward, but the cumulative reward of the whole trajectory, which is affected by the immediate reward and the transition (long-horizon rewards). However, this paper only considers the decomposition of immediate reward, ignoring the long-horizon causal relationship. For example, in the delayed reward setting (SMAC), the agents only receive the reward at the last timestep, the credit assignment of the whole trajectory in MACCA is only related to the state and actions of the last timestep, which is unreasonable.
>
> **Answer W1:** Thank you for the question. In our understanding, the setting you mentioned is generally in the single agent setting, which usually refers to temporal credit assignment, like having a delayed or episodic reward and assigning to each time step. However, our work in this paper primarily addresses spatial credit assignment within a multi-agent context, which involves determining the distribution of team rewards among individual agents, a process central to accurately assessing each agent's contribution.
>
> Furthermore, MACCA's primary objective is to learn an individual reward function to generate the $r_t^i$, distinct from directly apportioning the team reward $R_t$, like using the Shapley value [1][2]. Under our understanding, the reward itself, whether the team reward or individual reward, should not consider the long-term (long-horizon) effect, whereas, under the CTDE paradigm, the individual rewards derived are then utilized by the critic to calculate the corresponding $q_t^i$, ensuring that long-term effects are taken into account.
>
> Regarding the sparse reward setting in SMAC, it indeed poses a more complex challenge, combining both spatial and temporal credit assignment aspects. While this forms an intriguing direction for future research, it falls beyond the scope of our current paper, which concentrates on spatial credit assignment in offline multi-agent environments.
>
> - [1] Wang, Jianhong, et al. "Shapley Q-value: A local reward approach to solve global reward games." Proceedings of the AAAI Conference on Artificial Intelligence. Vol. 34. No. 05. 2020.
> - [2] Wang, Jianhong, et al. "Shaq: Incorporating shapley value theory into multi-agent q-learning." Advances in Neural Information Processing Systems 35 (2022): 5941-5954.
>
> > **Weakness 2**: The states of the environments adopted by this paper are low-dimension vectors. It is hard to capture the causal relationship between image state and reward.
>
> **Answer W2:** Thanks for the question. This work focuses mainly on state-based tasks. 1) For the setting where states are not defined, such as image-like input, our work can be applied by employed in the latent state space, which requires learning a latent vector representation. 2) For a large number of dimensions of state, this would increase the complexity of the causal modelling and thus require more efficient causal discovery methods. 3) As an advantage, by learning the causal structure, we can constrain the optimization of the models over a small subspace of state and action, resulting in a lower requirement of parameters of the neural network. 4) considering that we wanted a fairer comparison with other baselins under this setting, we followed the same data format and level of dimension of states with [1][2][3][4]. Overall, the main focus of this work is to address the team rewards credit assignment problem under the offline setting. Therefore, we consider the standard and general state setting.
>
> - [1] Samvelyan, Mikayel, et al. "The starcraft multi-agent challenge." arXiv (2019).
> - [2] Fu, Justin, et al. "D4rl: Datasets for deep data-driven reinforcement learning." arXiv (2020).
> - [3] Pan, Ling, et al. "Plan better amid conservatism: Offline multi-agent reinforcement learning with actor rectification." ICML. PMLR, 2022.
> - [4] Yang, Yiqin, et al. "Believe what you see: Implicit constraint approach for offline multi-agent reinforcement learning." Advances in Neural Information Processing Systems 34 (2021): 10299-10312.

---

> ### Author Response · Authors · 2023-11-16
> **Response to Reviewer SGeQ (2/2)**
>
> > **Question 1**: The proposed reward decomposition mechanism is not specific to offline MARL. The results will be more convincing if MACCA achieves performance gain in the online MARL (SMAC tasks).
>
> **Answer Q1:** We appreciate your interest in the applicability of our MACCA framework to online MARL settings. Our decision to focus on offline credit assignment was driven by several key factors. Firstly, offline spatial credit assignment poses a more significant challenge compared to online settings due to restricted exploration opportunities and the inherent constraints of data distribution. When relying only on team rewards, bias in offline data or significant performance differences between agents can seriously affect the accuracy of the evaluation. In contrast to offline settings, online credit assignment algorithms like COMA or SQDDPG can iteratively refine their evaluations through continuous interaction with the environment. Therefore, the challenge that MACCA focuses on in offline environments is accurate agents' contribution based on offline data, which is crucial to understand the role of each agent and optimize overall team performance in scenarios with restricted exploration.
>
> Secondly, the learning of causal models, which is a central component of MACCA, benefits from the more balanced data distribution typically found in offline reinforcement learning. In these settings, datasets often derive from policies with similar performance levels, resulting in a relatively uniform data distribution. This contrasts with online settings, particularly on-policy methods, where there's no inherent experience replay buffer to facilitate causal learning. Off-policy methods may offer some advantages, but they also come with the issue of potential shifts in the distribution within the experience replay buffer. Furthermore, while MACCA could theoretically be adapted for online off-policy settings—possibly through pre-training or the establishment of an independent causal learning buffer—such adaptations are beyond the current scope of our paper. Our focus on the offline setting is deliberate, as it naturally complements the learning of our causal model.
>
> Furthermore, MACCA has demonstrated superior performance across three distinct environments. We believe that these results affirm the efficacy of MACCA in the offline setting and hope this addresses any concerns regarding our choice of focus and the potential of MACCA in different application scenarios.

---

> > ### Comment · Reviewer_SGeQ · 2023-11-19
> >
> > As you claimed, your solution (decomposing reward) addresses spatial credit assignment without considering temporal credit assignment. However, there is an easy solution which decomposes the return rather than the reward. Decomposing return is long-horizon and considers both temporal and spatial credit assignments. Why do you chose decomposing reward?
> >
> > Decomposing reward might cause severe problem in credit assignment. If an agent choses an action which leads to a high global reward but a low global return. In your solution, this agent is the best over all agents, however, it is actually the main cause of the failure.
> >
> > As you claimed, your solution is not suitable for online learning, so I cannot believe that the causal model is really effective. Decomposing reward can naturally reduce the estimated value, which will benefit the offline learning and might be the reason for performance gain.

---

> > > ### Author Response · Authors · 2023-11-20
> > >
> > > We appreciate your reply and provide the following responses to your questions.
> > >
> > > > As you claimed, your solution (decomposing reward) addresses spatial credit assignment without considering temporal credit assignment. However, there is an easy solution which decomposes the return rather than the reward. Decomposing return is long-horizon and considers both temporal and spatial credit assignments. Why do you chose decomposing reward?
> > >
> > > **Response**: We divide the problem into two parts; refer to the Eq (1). For temporal, return, and long-term considerations, our policy learning (Sec 4.3) is based on the return at each step. The policy learning will leverage the return to optimize multiple agents as a team. In all MACCA-based methods, we use Q-Value to guide policy learning, each agent who estimates $\hat{Q}^i\left(o^i, a^i\right)=E_\pi\left[\sum_{t=0}^{\infty} \gamma^t \hat{r}_ t^i\right]$ with the Bellman backup operator, where the $\sum_{t=0}^{\infty} \gamma^t \hat{r}_t^i$ is the return (accumulate individual reward). For the causal modelling (Sec 4.2), we use the team reward label given by the environment to learn the causal model between states, actions and rewards.
> > >
> > >
> > > > Decomposing reward might cause severe problem in credit assignment. If an agent choses an action which leads to a high global reward but a low global return. In your solution, this agent is the best over all agents, however, it is actually the main cause of the failure.
> > >
> > > **Response**: In the policy learning as described in Section 3 Dec-POMDP setting and Section 4.3 objective functions, the objective for each agent's policy (Eq.7/8/9) is to choose the action that has maximum return rather than the current reward. So, it will not occur if the agent chooses the action which leads to a high team reward but a low team return.
> > >
> > > > As you claimed, your solution is not suitable for online learning, so I cannot believe that the causal model is really effective. Decomposing reward can naturally reduce the estimated value, which will benefit the offline learning and might be the reason for performance gain.
> > >
> > > **Response**: We didn't claim that MACCA is not suitable for the online setting; in contrast, the proof of the identifiable is also suitable for the online setting. However, we emphasize in the paper and in our previous response that even though it is a good research direction, it is out of our research scope, and we claimed that the offline spatial credit assignment problem is more challenging than the online setting. For the performance gain, we tested MACCA in three distinct environments, and all show SOTA performance compared to the algorithms using conservative Q-value estimations; according to the expirements, ablation test and the improvement table below, we believe that MACCA improves the performance due to causal modeling.
> > >
> > >
> > > | Env/Algo| MACCA-CQL | MACCA-OMAR| MACCA-ICQ|
> > > | -------- | -------- | -------- |-------- |
> > > | MPE     | +847.9%   | +62.0%      | +237.8%     |
> > > | MA-Mujoco| +24.0%   | +44.6%      | +92.3%      |
> > > | SMAC    | +788.5%   | +154.17%    | +271.77%      |

---

> > > > ### Author Response · Authors · 2023-11-22
> > > >
> > > > Dear Reviewer SGeQ, we hope our answers and results above have addressed your concerns. If so, we would greatly appreciate it if you could reconsider your score. Please let us know if there are any other questions.

---

> > > > > ### Comment · Reviewer_SGeQ · 2023-11-22
> > > > >
> > > > > >Response 1
> > > > >
> > > > > Apparently, policy learning is guided by return, but the return is the sum of decomposed individual rewards, which is different from the decomposed global return. This is what I mean the difference between your solution and the long-horizon solution I proposed. And you did not answer this question.
> > > > >
> > > > > Let us discuss a toy case. There are two agents (1 and 2) and an enemy. If the enemy is attacked four times, it is killed and the agents receive a global 10. Otherwise, the reward is 0. In an episode, agent 1 attacks the enemy in the first three timesteps, and agent 2 attacks the enemy in the last timestep and kills the enemy. Could you analyze the temporal and spatial reward assignment under your solution (decomposing the rewards and then computing individual return) and the long-horizon solution (computing global return and decomposing it). We know agent 1 contributes more than agent 2. But you solution will give agent 2 more reward.
> > > > >
> > > > > > Response 2 : So, it will not occur if the agent chooses the action which leads to a high team reward but a low team return.
> > > > >
> > > > > In the training process, the agents will chose random actions for exploration. So this situation will occur in the training process and the agents will receive a wrong reward assignment under your solution in that case, which will mislead the training.
> > > > >
> > > > > > Response 3
> > > > >
> > > > > If the proposed credit assignment method really effective, it should work in both offline and online settings. Performing online experiments is the only way to demonstrate the effectiveness in online setting, since you claim that online setting is easier than offline setting. And online experiments can greatly improve the significance of your paper.

---

> ### Author Response · Authors · 2023-11-22
>
> Thank you for your reply. Before answering your question, we would like to clarify the composition of the reward function in the offline dataset we use.
>
> **Team Reward Format in Offline Dataset**: We would like to describe the offline datasets more, since it seems there might be a misunderstanding about the nature of the offline RL setting and the dataset we are using.
>
> - **SMAC environment reward (team reward)**: the team reward is generated based on the hit-point damage inflicted on enemy units and the number of enemy units eliminated at each timestep. To illustrate, consider the scenario in the 5m_vs_6m map within the Medium dataset. From timestep 0 to 10, the team rewards are as follows: (0, 0, 0, 0.9057, 0.2264,0.6792, 0.7170, 0.679,0.2264, 0.4528). Here the team reward is **0** at timesteps zero to three, **0.9057** at timestep four, **0.2264** at timestep five, **0.6792** at timestep six, **0.7170** at timestep seven, **0.679** at timestep eight, **0.2264** at timestep nine, and **0.4528** at timestep ten. This demonstrates that team rewards are evaluated based on the team's performance against the opponent at each timestep, rather than cumulatively.
>
> - **MA-MuJoCo enviornment reward (team reward)**: the reward function is formulated as $\frac{\Delta x}{\Delta t}+0.1 \alpha$. Here, $\Delta x$ denotes the change in the overall position or distance, $\Delta t$ signifies the change in time and $\alpha$ represents an action regularization term.
>
> - **MPE environment reward (team reward)**: For example, in Cooperative Navigation (CN), all agents are globally rewarded based on how far the closest agent is to each landmark (sum of the minimum distances) for each timestep. The team reward are penalized if some agents collide with other agents (-1 for each collision).
>
> > Q1: Apparently, policy learning is guided by return, but the return is the sum of decomposed individual rewards, which is different from the decomposed global return. This is what I mean the difference between your solution and the long-horizon solution I proposed.
>
> **Response**:  We are not arguing against your solution: your approach is indeed a valid one. However, it might require a reconstruction of the environmental dataset rewards; therefore, it is not suitable for our offline setting. Since you raised concerns about our method not considering return and long-horizon effects, we want to clarify that our solution does take return into account: the policy learning will learn a policy that maximizes the long-term return. In conclusion, our approach considers the setting of currently available offline benchmark datasets, where each time step represents an instantaneous team reward rather than an accumulated reward. Then we have adopted the proposed approach.
>
> > Q2: Analysis of toy case & Q3: In the training process, the agents will chose random actions for exploration. So this situation will occur in the training process and the agents will receive a wrong reward assignment
>
> **Response**:  Thank you for providing this example. According to the **Team Reward Format** we provide, unlike the scenario in this toy example, the team reward at each time step reflects the collective performance of the team at the current timestep rather than being cumulative. In your example:
>
> - At timestep=1, Agent 1 attacks, env reward (i.e., team reward from the dataset) = 1. Agent 1 would receive a higher reward, let's say 0.99, and Agent 2 would receive 0.01.
> - At timestep=2, Agent 1 attacks, env reward=1. Again, Agent 1 would receive 0.99, and Agent 2 would receive 0.01.
> - At timestep=3, Agent 1 attacks, env reward=1. Once more, Agent 1 would receive 0.99, and Agent 2 would receive 0.01.
> - At timestep=4, Agent 2 attacks, env reward=1. Now, Agent 2 would receive a higher reward, let's say 0.99, and Agent 1 would receive 0.01.
>
> It's important to note that at timestep=4, the env reward is not 4 but 1, as provided directly by the environmental data. Our algorithm establishes a relationship between team reward and individual agent contributions, allowing it to better infer that Agent 1 should receive more rewards overall. Thank you for presenting this example.
>
> > Q4: It should work in both offline and online settings. Performing online experiments is the only way to demonstrate the effectiveness in online setting.
>
> **Response**:  Once again, we agree with your suggestion to try it online. However, what we would like to emphasize is that this paper, as indicated by our title, is focused on an offline setting. We opted for an offline setting because, as mentioned earlier, challenges in environments like the online SMAC environment have largely been addressed. Furthermore, tackling offline scenarios presents greater difficulty and room for improvement, which is why we conducted our experiments in an offline setting. If there is a desire to explore online scenarios, that could potentially be the subject of a separate paper. Nevertheless, given tha

---

### Official Review · Reviewer_XKou · 2023-10-30

**Soundness:** 3 good
**Presentation:** 2 fair
**Contribution:** 2 fair
**Rating:** 5
**Confidence:** 5

**Summary:**

This paper looks into the problem of credit assignment for individual agents in a shared environment, with the potential impact of other issues such as partial observability and emergent behavior. Specifically in offline multi-agent reinforcement learning (MARL) settings, the diversity in different distributions of data complicates the task of assigning individual credits. To address this problem, the paper proposes a framework named Multi-Agent Causal Credit Assignment (MACCA). After discussing the related work in offline MARL and multi-agent credit assignment and the preliminaries of the method, the paper breaks down the MACCA framework into offline data generation, causal model learning, and policy learning with assigned individual rewards. The estimates of the observation and reward produced by the agents in the policy learning phase are fed into the policy network for generating the next-state actions. The authors conducted experiments in various environments and compared their method with the other baselines that follow the centralized training with decentralized execution (CTDE) and independent learning paradigms. They also carried out ablation studies to evaluate the interpretability and efficiency of their approach.

**Strengths:**

The methodology in this paper is written in an organized manner. At the start of section 4, the two major components (causal and policy models) are described. The overall objective is defined and then elaborated in the subsections. It is very obvious that the method consists of generative process, causal model learning phase, and policy optimization by reading this section. In the experiments, a sufficient number of baselines are applied for performance comparison and most of them are relatively new. Three different MARL testbeds are used for evaluation, which shows the generalization capability of the method. Careful ablation studies are performed in order to clarify the impacts of causal structure, ground truth individual reward, and causal graph setting. Furthermore, the math used here is formal and clear, improving the soundness of this work. The proof of identifiability is correct and a good supplement to the paper.

**Weaknesses:**

The overall writing quality needs to be improved. There are some grammatical errors, and the details will follow.

Although the evaluation of MACCA has been conducted in multiple benchmarks to demonstrate its generalizability, it is questionable whether the method has top performance in all of the environments that belong to a specific benchmark. For example, the SMAC benchmark has more than 20 original battle scenarios, including a few Super Hard challenges. However, only three of the maps, including a single Super Hard challenge, are shown in the results.

Regarding the MACCA architecture, clearly, the policy learning part is crucial for the model. However, this part is mostly taken from previously established methods such as I-CQL, OMAR and MA-ICQ, especially the term $J_{\pi}$. Also, after looking at the experiments in different environments, it is not clear whether MACCA-CQL, MACCA-OMAR, or MACCA-ICQ has the SOTA performance overall.

The code for the proposed method is not included in the submission. The disclosed information about the hyper-parameters and environmental settings is limited.

A few more points worth mentioning:

- The figure or the algorithm in Appendix D can be moved to the main paper to improve the clarity of the description for MACCA.

- In section 3, when defining the Dec-POMDP, an additional $\Omega$ is included in the tuple as it denotes the joint observation space. Then the observation function is expressed as $\mathcal{O}(s, i) : \mathcal{S} \times \mathcal{A} \rightarrow \Omega$.

- In section 4.1: Define $D_s$ and $D_a$ as the {numbers} of dimentions of ... The {masks}... are vectors and ...

- In section 4.2: $\psi_r$ is {used} for {approximating} ...

- In section 4.3: each agent's state-action-id {tuple} ...

- In the first paragraph of section 5: ... variants of credit assignment method using {Shapley} value, ...

**Questions:**

-In section 4.1, are the masks in the expression for the reward learned or manually set?

-There is a long and confusing sentence in section 4.2: "Its primary objective is to ... reduce the number of features that a given depends on, ... mitigates the risk of overfitting." What does "reduce the number of features that a given depends on" mean?

-Is the hyper-parameter $h$ learned?

-Can you mention how many independent tasks do the Multi-agent Particle Environment and the Multi-agent MuJoCo have, respectively?

-In section 5.3 you mentioned that "It is important to note that our method is not highly sensitive to the hyperparameters despite using them to control the learned causal structure." Can you justify this argument?

-You showed the visualizations of two causal structures in MPE. Have you done similar work in MA-Mujoco and SMAC?

-In the sub-section "Visualization of Causal Structure." in 5.3, what do "S2R" and "A2R" stand for?

**Details Of Ethics Concerns:**

There is no ethics concern as far as I can tell.

---

> ### Author Response · Authors · 2023-11-16
> **Response to Reviewer XKou (1/2)**
>
> We thank the reviewer for the constructive comments and provide our point-wise response below.
>
> > **Weakness 1**: The overall writing quality needs to be improved. There are some grammatical errors, and the details will follow.
>
> **Answer W1:** Thanks for pointing it out. We will revise our paper carefully in the revision.
>
> > **Weakness 2**: Although the evaluation of MACCA has been conducted in multiple benchmarks to demonstrate its generalizability, it is questionable whether the method has top performance in all of the environments that belong to a specific benchmark. For example, the SMAC benchmark has more than 20 original battle scenarios, including a few Super Hard challenges. However, only three of the maps, including a single Super Hard challenge, are shown in the results.
>
> **Answer W2:** Thank you for your concern. Our paper mentions that the primary reason for choosing the SMAC benchmark is to demonstrate performance in larger-scale scenarios (Section 5.2). Accordingly, we focused on maps that are challenging and involve more than five agents. However, considering your concern and acknowledging that the extensive experimentation may further substantiate MACCA's SOTA performance, we will include additional Super Hard map in our next revision.
>
> > **Weakness 3**: Regarding the MACCA architecture, clearly, the policy learning part is crucial for the model. However, this part is mostly taken from previously established methods such as I-CQL, OMAR and MA-ICQ, especially the term $J_\pi$. Also, after looking at the experiments in different environments, it is not clear whether MACCA-CQL, MACCA-OMAR, or MACCA-ICQ has the SOTA performance overall.
>
> **Answer W3:** Thank you for highlighting this aspect of the MACCA architecture. Firstly, it's important to note that MACCA is designed to be a plug-and-play method, capable of being seamlessly integrated with other offline MARL frameworks. This adaptability is, in fact, one of MACCA's strengths, as it allows for versatile applications across a range of existing methods.
>
> | Env/Algo| MACCA-CQL | MACCA-OMAR| MACCA-ICQ|
> | -------- | -------- | -------- |-------- |
> | MPE     | +847.9%   | +62.0%      | +237.8%     |
> | MA-Mujoco| +24.0%   | +44.6%      | +92.3%      |
> | SMAC    | +788.5%   | +154.17%    | +271.77%      |
>
>
> Secondly, regarding the performance of MACCA in different environments, we provide detailed data in the paper to prove its efficacy. Here, we summarize the results in the table above, including the average percentage improvement of all MACCA-based methods in these three environments. According to the results presented in our paper, MACCA-based algorithms consistently meet or exceed SOTA performance in all test environments and maps. This is demonstrated by the bold entries in Tables 1 and 2, which highlight the best performance in the experiments.
>
>
>
> > **Weakness 4**: The code for the proposed method is not included in the submission. The disclosed information about the hyper-parameters and environmental settings is limited.
>
> **Answer W4:** We appreciate your feedback. The description of hyperparameters is in Appendix D, including the model structure and specific hyperparameters. The environment settings are in Appendix C, including the configuration and data set generation methods of three different environments. Also, to enhance clarity and reproducibility, we have uploaded the code for our proposed method to an anonymous repository, accessible at: https://anonymous.4open.science/r/MACCA_ICLR/.
>
> > **Question 1** In Section 4.1, are the masks in the expression for the reward learned or manually set?
>
> **Answer Q1:** The causal masks are learnable and predicted by networks, enjoying the advantage of addressing complex dynamic settings in MARL, described in equation 3, Section 4.2.
>
> > **Question 2** There is a long and confusing sentence in Section 4.2: "Its primary objective is to ... reduce the number of features that a given depends on, ... mitigates the risk of overfitting." What does "reduce the number of features that a given depends on" mean?
>
> **Answer Q2:** We will clarify this in the future version: the $L_{\mathrm{reg}}$ can reduce the difficulty of estimation of the reward function since it can clear the redundant features during training by reducing the sparsity of the causal structure $\mathcal{S}_{s r}$ and $\mathcal{S}_{a r}$ (defined in Section 5.3), in order to mitigate the risk of overfitting.
>
>
> > **Question 3** Is the hyper-parameter $h$ learned?
>
> **Answer Q3:** No, the design specifics of hyperparameter $h$ is elaborated in Section 4.3 Individual Rewards Assignment, and the value of $h$ is also elaborately in Appendix D.3 table A2, we set $h$ in all environments to 0.1.

---

> ### Author Response · Authors · 2023-11-16
> **Response to Reviewer XKou (2/2)**
>
> > **Question 4** Can you mention how many independent tasks do the Multi-agent Particle Environment and the Multi-agent MuJoCo have, respectively?
>
> **Answer Q4:** Yes, as described in Appendix C, there are three scenarios (PP, CN and World) in the Multi-agent Particle Environment and one scenario (Half-Cheetah) in Multi-agent MuJoCo.
>
> > **Question 5** In Section 5.3, you mentioned that "It is important to note that our method is not highly sensitive to the hyperparameters despite using them to control the learned causal structure." Can you justify this argument?
>
> **Answer Q5:** Thank you for the question. To investigate the impact of the hyperparameters, we refer to the results presented in Section 5.3 Table 4. This ablation study discusses how different values of \(\lambda_1\) affect the sparsity of the learned causal structure. The results indicate that while varying $\lambda_1$ leads to changes in the sparsity of the causal structure, the impact on the convergence of policy learning is relatively minimal.
>
> | $\lambda_2$ / t | 1e4   | 5e4  | 1e5 | 2e5 |
> | -------- | -------- | -------- | -------- | -------- |
> | 0     |    17.4 ± 15.2(0.98) |  93.1 ± 6.4  (1.0)  | 105 ± 3.5  (1.0)  | 107.7 ± 10.2 (1.0)   |
> | 0.007    |  19.9 ± 12.4 (0.88) | 90.2 ± 7.1 (1.0)   |  108.8 ± 4.0 (1.0) | **111.7 ± 4.3** (1.0)    |
> | 0.5    | 13.3 ± 11.1 (0.68)    | 100.5 ± 14.0  (0.84)   | 102.9 ± 16.4 (0.87)  | 108.4 ±  6.4 (0.98) |
> | 5.0    | 2.3 ± 9.8  (0.0)    | -1.3 ± 25.4   (0.34)  | 70.4 ± 18.0  (0.62)   | 100.1 ±  7.4  (0.75)  |
>
> Additionally, we add an additional ablation study on $\lambda_2$ as shown in the above table, which presents the mean and the standard variance of the average normalized score with diverse $\lambda_2$ in the MPE-CN task. The value in brackets is the sparsity rate $\mathcal{S}_ {ar}$ of $\hat{\boldsymbol{c}}_ t^{i, a \rightarrow r}$, whose definition can be found in Section 5.3. For all values of $\lambda_2$, the sparsity rate $\mathcal{S}_ {a r}$ consistently begins from zero. Over time, there is a discernible increase in $\mathcal{S}_{a r}$, and the convergence rate slows down with the increase of $\lambda_2$. And, the convergence rate to an elevated sparsity rate decelerates with the increase of $\lambda_2$. This pattern intimates that higher $\lambda_2$ values engender a more measured modulation in the causal impact exerted by actions on individual rewards. More importantly, despite the variation in $\lambda_2$ values, the average normalized scores converged towards a similar range.
>
>
>
>
>
> > **Question 6** You showed the visualizations of two causal structures in MPE. Have you done similar work in MA-Mujoco and SMAC?
>
> **Answer Q6:** Thank you for the question. The primary reason we focused on visualizations in MPE is due to its comparatively smaller state and action spaces relative to MA-Mujoco and SMAC. This smaller scale facilitates a more straightforward presentation and more accessible human analysis. Taking into account your suggestions regarding extending these efforts to MA-Mujoco and SMAC, we recognize its value and will update it in future revision.
>
> > **Question 7** In the sub-section "Visualization of Causal Structure." in 5.3, what do "S2R" and "A2R" stand for?
>
> **Answer Q7:** Thanks for pointing out these typos, "S2R" refers to Figure 2.a, which is "Causal Structure: States to Individual Rewards", and "A2R" refers to Figure 2.b, which is "Causal Structure: Individual Reward Actions".

---

> > ### Author Response · Authors · 2023-11-20
> >
> > Dear reviewer, we appreciate your valuable feedback and constructive comments. Since there are only three days left in the rebuttal process, we want to know if our response addresses your concerns. We are willing to engage in further discussions and welcome any additional suggestions that may help improve the quality of the paper.

---

> > > ### Author Response · Authors · 2023-11-22
> > >
> > > Dear Reviewer XKou, we hope our answers and results above have addressed your concerns. If so, we would greatly appreciate it if you could reconsider your score. Please let us know if there are any other questions.

---

> > > > ### Comment · Reviewer_XKou · 2023-11-22
> > > >
> > > > Dear authors, unfortunately, I do not think the responses have adequately addressed the existing weaknesses so that a raise of the score can be given. I did not see any new submissions, although you have mentioned you would present newer revisions. Still, I would like to thank you for the clarifications.

---

> > > > > ### Author Response · Authors · 2023-11-23
> > > > >
> > > > > Dear Reviewer XKou,
> > > > >
> > > > > Thank you for your feedback. We submitted a new revision of our paper, addressing the concerns you raised. Due to the tight timeline, the following key updates have been made in this latest version (the modified parts are highlighted in **orange** ):
> > > > >
> > > > > 1. **Enhanced Writing and Proofreading:** We have meticulously revised the manuscript to improve clarity, corrected grammatical errors, and eliminated typographical mistakes.
> > > > >
> > > > > 2. **Inclusion of $(\lambda_2\)$ Ablation Study:** An extensive ablation study focusing on $\lambda_2\$ and its detailed analysis have been added to the appendix, providing deeper insights into its impact on our model.
> > > > >
> > > > > 3. **Exploration of MACCA in Online Settings:** We have expanded our analysis to include MACCA's performance in online environments. Specifically, we combined MACCA with the off-policy algorithm IQL and observed notable improvements in SMAC tasks. This showcases MACCA's adaptability and effectiveness in diverse settings.
> > > > >
> > > > > 4. **Addition of a Super-Hard Map in SMAC Environment:** To further demonstrate the robustness of MACCA, we included a new Super-Hard map, MMM2, in the SMAC environment. Even in this more challenging scenario, MACCA- based algorithms continued to exhibit state-of-the-art performance.
> > > > >
> > > > > We hope these updates comprehensively address the points raised and enhance the quality of the paper. Please let us know if there are any other questions.

---

### Official Review · Reviewer_jLy1 · 2023-10-31

**Soundness:** 3 good
**Presentation:** 3 good
**Contribution:** 3 good
**Rating:** 6
**Confidence:** 3

**Summary:**

This paper studies offline multi-agent reinforcement learning. It proposes to learn the causal structure between states and actions and the team reward from the offline dataset with supervised learning to tackle the credit-assignment problem. Experiments are conducted to demonstrate the effectiveness of the algorithm.

**Strengths:**

1. The empirical performance looks compelling.
2. The idea of extracting the causal structure behind the team reward is interesting.
3. This paper is clearly written and easy to follow.

**Weaknesses:**

1. The presumed data-generating process is restrictive.

Minor Mistake:
There is a 'shaply' on the seventh line of the first paragraph of Section 5

**Questions:**

1. Have the authors ever considered extending the algorithm to scenarios where the team reward cannot be decomposed as the sum of individual rewards? What can be the possible solution?

---

> ### Author Response · Authors · 2023-11-16
> **Response to Reviewer jLy1**
>
> We thank the reviewer for your positive support. Below, we provide a point-wise response to your questions.
>
> > **Weakness 1**: The presumed data-generating process is restrictive.
>
> **Answer W1:**
>
> Thank you for the question. We understand your concern regarding the expressive capability of our linear model in MACCA. Firstly, it is essential to emphasize that MACCA is adaptable for some specific nonlinear team reward functions. For instance, in cases where the team reward $R$ is defined as either the maximum or minimum of individual rewards $r_t^i$ in the team, the identifiability of individual rewards under this setting is retained, and our theoretical proofs remain valid. Secondly, our linear configuration does not directly impact the training of the network. Although our model assumes the team reward $R$ as the sum of individual rewards, the primary output of the $\psi_r$ is the estimated individual rewards $\hat{r}_t^i$, not directly generating the estimated team reward $\hat{R}_t$. Finally, team reward can be modelled as linear in many scenarios, particularly in cooperative games. A pertinent example is urban traffic management, where multiple agents control traffic signals at various intersections. Here, the traffic flow at each intersection and the overall throughput of the system exhibit a linear relationship. By aggregating the contributions from each intersection, the effectiveness of the entire traffic management system can be effectively quantified. Therefore, the linear model is sufficient so far, and we are considering discussing more complex settings and providing corresponding solutions in future work.
>
> > **Questions 1**: Have the authors ever considered extending the algorithm to scenarios where the team reward cannot be decomposed as the sum of individual rewards? What can be the possible solution?
>
> **Answer Q1:**
> Thank you for the question. Yes, we will consider extending to such non-sum scenarios in future work. The possible direction is to use some nonlinear integration function to generate the team reward instead of the sum function.

---

> > ### Author Response · Authors · 2023-11-20
> >
> > Dear reviewer, we appreciate your valuable feedback and constructive comments. Since there are only three days left in the rebuttal process, we want to know if our response addresses your concerns. We are willing to engage in further discussions and welcome any additional suggestions that may help improve the quality of the paper.

---

> > > ### Comment · Reviewer_jLy1 · 2023-11-21
> > >
> > > Thank you for the detailed explanations! I understand that linear functions work in some generality and I keep my recommendation for acceptance.

---

> > > > ### Author Response · Authors · 2023-11-22
> > > >
> > > > Thank you for your positive feedback on our work; your further consideration for a higher score, acknowledging its broader impact, would be greatly appreciated.

---

### Official Review · Reviewer_9GYs · 2023-11-01

**Soundness:** 3 good
**Presentation:** 3 good
**Contribution:** 3 good
**Rating:** 6
**Confidence:** 4

**Summary:**

This paper presents the MACCA algorithm, which describes the generation process as a dynamic Bayesian network, capturing the relationships among variables, states, actions, and rewards in the environment. By analyzing the causal relationships of agent rewards, it learns the contribution of each agent, addressing the credit assignment problem in offline multi-agent reinforcement learning. Specifically, MACCA employs the Bayesian network $G$ to construct the causal relationships among states, actions, and individual rewards. It models the relationship of state $\boldsymbol{s}$ and action $\boldsymbol{a}$ to individual reward $r_t^i$ through masks $C^{i, s \rightarrow r}$ and $C^{i, a \rightarrow r}$ MACCA's loss integrates losses from both the causal model and the policy model. MACCA's superiority is demonstrated on offline datasets from MPE, MA-MuJoCo, and SMAC.

**Strengths:**

1. The paper is well-written and easy to understand.  The original contributions are highlighted clearly.
2. This paper provides a thorough and complete set of theoretical proofs. The proofs provided are clear, rigorous.
3. Experiments/ablations are abundant, and experimental results are convincing.

**Weaknesses:**

1. While MACCA establishes causal relationships among states, actions, and individual rewards in offline datasets, this "causal relationship" doesn't seem to have a strong correlation with offline multi-agent reinforcement learning. It appears that this "causal relationship" might also be applicable in online reinforcement learning，rather than being specifically designed for offline multi-agent reinforcement learning.
 2. The design specifics of $c^{i, s \rightarrow r}$ and $c^{i, a \rightarrow r}$ are not elaborated upon, and the particular networks used are not clearly mentioned.
 3. There isn't much explanation regarding the setting of the hyperparameter $h$, nor is there any mention of whether the value of $h$ remains consistent across different offline environments.
 4. Concrete code has not been provided.

**Questions:**

1. Is the MACCA algorithm applicable to online reinforcement learning? Because it seems that in online reinforcement learning, causal relationships can also be applied, and the causal relationships among states, actions, and individual rewards can be constructed through Bayesian networks. Why is it emphasized that the MACCA algorithm is primarily for offline environments?
2. The paper's explanation regarding the setting and values of the hyperparameter $h$ seems to be unclear. Is the value of $h$ set the same across all offline datasets?
3.  In the ablation study section, the impact of $\lambda_1$ on the causal structure, specifically its influence on the state's effect on individual rewards, was explored. However, has the impact of $\lambda_2$ on the causal structure been considered? I didn't see this part being researched in the paper.
4.  In the related work section on multi-agent credit assignment, it seems that recent approaches in value decomposition, such as wqmix, qplex, resq, etc., have not been considered. These methods tackle the credit assignment challenge among agents using value decomposition techniques and, from what I understand, they have demonstrated commendable performance. Have you considered applying the methods from wqmix, qplex, resq, etc., to offline multi-agent environments?

---

> ### Author Response · Authors · 2023-11-16
> **Response to Reviewer 9GYs (1/2)**
>
> Thank you very much for your positive feedback. We sincerely appreciate your positive support and constructive comments. Below, we provide a point-wise response to your questions.
>
> > **Weaknesses 1 and Question 1**: While MACCA establishes causal relationships among states, actions, and individual rewards in offline datasets, this "causal relationship" doesn't seem to have a strong correlation with offline multi-agent reinforcement learning. It appears that this "causal relationship" might also be applicable in online reinforcement learning，rather than being specifically designed for offline multi-agent reinforcement learning.
>
> **Response to W1&Q1:**  We appreciate your interest in the potential application of our MACCA framework to online MARL settings. Our choice to focus on the offline credit assignment setting was driven by several primary considerations.
>
> Firstly, the restricted exploration in offline settings poses unique challenges to this credit assignment problem. When relying only on team rewards, bias in offline data or significant performance differences between agents can seriously affect the accuracy of the evaluation. In contrast to offline settings, online credit assignment algorithms like COMA or SQDDPG can iteratively refine their evaluations through continuous interaction with the environment. Therefore, the challenge that MACCA focuses on in offline environments is accurate agents' contribution based on offline data, which is crucial to understanding the role of each agent and optimising overall team performance in scenarios with restricted exploration.
>
> Secondly, an essential aspect of MACCA is the training of causal models, which significantly benefits from the stable and consistent data distribution typically found in offline reinforcement learning environments. In offline settings, data often comes from policies with comparable performance levels, leading to a more uniform distribution. This stability is advantageous for causal learning, as it reduces the variability and potential distribution shifts that can occur in online settings, particularly in online and on-policy scenarios where an experience replay buffer is not utilized. Off-policy methods in online settings may offer some mitigation against this issue, but they are not without their challenges, such as potential shifts in the experience replay buffer's distribution. Given these considerations, our focus on offline settings was a deliberate choice, aligning well with the objectives and strengths of the MACCA framework.
>
> Furthermore, MACCA has demonstrated superior performance across three distinct environments. We believe that these results affirm the efficacy of MACCA in the offline setting and hope this addresses any concerns regarding our choice of focus and the potential of MACCA in different application scenarios.
>
> > **Weaknesses 2**: The design specifics of $c^{i, s \rightarrow r}$ and $c^{i, a \rightarrow r}$ are not elaborated upon, and the particular networks used are not clearly mentioned.
>
> **Answer w2:** Apologies for any confusion. In fact, we have introduced the design specifics of $c^{i, s \rightarrow r}$ and $c^{i, a \rightarrow r}$ in Section 4.2 and describe the networks used in Appendix D.2. The masks $\boldsymbol{c}^{i, \boldsymbol{s} \rightarrow r} \in\{0,1\}^{D_s}$ and $\boldsymbol{c}^{i, \boldsymbol{a} \rightarrow r} \in\{0,1\}^{D_a}$ in equation 2 denote as the binary masks in which control if a specific dimension of the state $s$ and action $a$ impact the individual reward $r_t^i$, separately. To estimate them, we use $\psi_g^{s \rightarrow r}$ and $\psi_g^{a \rightarrow r}$ in equation 3, the network we used share the same architecture but the parameters. It includes three fully-connected layers with a hidden size of 256, followed by an output layer with a scalar output. Each hidden layer is activated using the rectified linear unit (ReLU) activation function.
>
> > **Weaknesses 3 and Question 2**: There isn't much explanation regarding the setting of the hyperparameter $h$, nor is there any mention of whether the value of $h$ remains consistent across different offline environments.
>
> **Answer W3&Q2:** Thank you for your query. The design specifics of hyperparameter $h$ is elaborated in Section 4.3 Individual Rewards Assignment, and the value of $h$ is also elaborately in Appendix D.3 Table A2.
>
> The hyperparameter $h$ is a threshold to determine the existence of the causal edges after we get the scalar output of $\psi_g^{s \rightarrow r}$ or $\psi_g^{a \rightarrow r}$ in equation 3. During the inference, i.e., predicting the individual rewards for policy optimization, the binary masks to capture the causal structure are obtained by $\boldsymbol{c} =I(ReLU(o)>h)$ where $o$ is the output of the last layer of $\psi_g^{s \rightarrow r}$ or $\psi_g^{a \rightarrow r}$, and $I$ is indicator function. The value of the $h$ is invariant across different experiments, which is $0.1$, as shown in Table A2.

---

> ### Author Response · Authors · 2023-11-16
> **Response to Reviewer 9GYs (2/2)**
>
> > **Weaknesses 4**: Concrete code has not been provided.
>
> **Answer w4:** We appreciate your feedback and have uploaded the code to anonymous git: https://anonymous.4open.science/r/MACCA_ICLR/ to improve clarity and support reproducibility.
>
>
> > **Question Q3**:In the ablation study section, the impact of $\lambda_1$ on the causal structure, specifically its influence on the state's effect on individual rewards, was explored. However, has the impact of $\lambda_2$ on the causal structure been considered? I didn't see this part being researched in the paper.
>
> **Answer 3:** Thank you for your question. We have conducted additional experiments on $\lambda_2$ and show the results in the table below.
>
> | $\lambda_2$ / t | 1e4   | 5e4  | 1e5 | 2e5 |
> | -------- | -------- | -------- | -------- | -------- |
> | 0     |    17.4 ± 15.2(0.98) |  93.1 ± 6.4  (1.0)  | 105 ± 3.5  (1.0)  | 107.7 ± 10.2 (1.0)   |
> | 0.007    |  19.9 ± 12.4 (0.88) | 90.2 ± 7.1 (1.0)   |  108.8 ± 4.0 (1.0) | **111.7 ± 4.3** (1.0)    |
> | 0.5    | 13.3 ± 11.1 (0.68)    | 100.5 ± 14.0  (0.84)   | 102.9 ± 16.4 (0.87)  | 108.4 ±  6.4 (0.98) |
> | 5.0    | 2.3 ± 9.8  (0.0)    | -1.3 ± 25.4   (0.34)  | 70.4 ± 18.0  (0.62)   | 100.1 ±  7.4  (0.75)  |
>
> This table shows the mean and the standard variance of the average normalized score with diverse $\lambda_2$ in the MPE-CN task. The value in brackets is the sparsity rate $\mathcal{S}_ {ar}$ of $\hat{\boldsymbol{c}}_ t^{i, a \rightarrow r}$, whose definition can be found in Section 5.3. For all values of $\lambda_2$, the sparsity rate $\mathcal{S}_ {a r}$ consistently begins from zero. Over time, there is a discernible increase in $\mathcal{S}_{a r}$, and the convergence speed slows down with the increase of $\lambda_2$. This pattern intimates that higher $\lambda_2$ values engender a more measured modulation in the causal impact exerted by actions on individual rewards. Furthermore, despite the variation in $\lambda_2$ values, the average normalized scores across different $\lambda_2$ settings eventually converge towards a similar level.
>
> > **Question 4**: In the related work section on multi-agent credit assignment, it seems that recent approaches in value decomposition, such as wqmix, qplex, resq, etc., have not been considered. These methods tackle the credit assignment challenge among agents using value decomposition techniques and, from what I understand, they have demonstrated commendable performance. Have you considered applying the methods from wqmix, qplex, resq, etc., to offline multi-agent environments?
>
> **Answer Q4:** Thank you for the question. As reflected in Table 3, we have conducted a comprehensive comparison with SOTA online off-policy credit assignment baselines (including QMIX, SHAQ, and SQDDPG) and their combination with offline learning algorithms to maintain a fair and unbiased comparison. We noticed that the plethora of QMIX variants that employ diverse mixing networks for value decomposition, their application in offline settings is indeed worth considering for a more exhaustive comparison. Nevertheless, our comparative analysis primarily concentrates on explicit credit assignment methods. Upon a thorough evaluation of both the performance and robustness of these implicit credit assignment methods in online scenarios, we opted to utilize QMIX as a representative model for all value decomposition approaches.

---

> > ### Author Response · Authors · 2023-11-20
> >
> > Dear reviewer, we appreciate your valuable feedback and positive comments. Since there are only three days left in the rebuttal process, we want to know if our response addresses your concerns. We are willing to engage in further discussions and welcome any additional suggestions that may help improve the quality of the paper.

---

> > > ### Comment · Reviewer_9GYs · 2023-11-22
> > >
> > > I would like to thank the authors for their feedback.  I have read other reviewers' comments and the author's replies. The authors' answers have resolved most of my queries. I will keep my score. Thanks.

---

> > > > ### Author Response · Authors · 2023-11-22
> > > >
> > > > Thank you for your thoughtful consideration and understanding of our responses. We are pleased to have addressed your queries satisfactorily. In light of this, we kindly hope you might reconsider the possibility of a higher score, reflecting the resolved concerns and the merit of our work.

---

### Meta-Review · Area_Chair_QG5N · 2023-12-06

**Metareview:**

This paper proposes MACCA, a reward decomposition mechanism for offline MARL. MACCA factorizes the shared reward into individual rewards according to the causal relationship among states, actions, and rewards. Then, each agent learns to optimize the individual rewards by existing offline methods. The experiments are performed on offline datasets of MPE, MA-MuJoCo, and SMAC and show better performance over baselines.

The main weaknesses are as follows.

1. Decomposing reward is rather a limited method. It cannot handle sparse reward settings. Moreover, as in MACCA each agent optimizes the cumulative decomposed rewards individually, how all agents jointly learn towards optimizing the original share return. There is no evidence or theoretical support for this.

2. As the proposed method is not specifically tailored for offline RL. It is necessary to verify whether the proposed method works for online settings. However, current experimental results (3m and MMM2 in SMAC) make the effectiveness of the proposed method in online settings inconclusive, since training steps are limited, not usually 2M steps in SMAC.

These weaknesses make this paper not ready for publication.

**Justification For Why Not Higher Score:**

The main weaknesses are not addressed during the rebuttal.

**Justification For Why Not Lower Score:**

N/A

---

### Decision · Program_Chairs · 2024-01-16

Reject